

# Channel concavity controls plan-form complexity of branching drainage networks

Liran Goren[1] and Eitan Shelef[2]

[1]The Department of Earth and Environmental Sciences, Ben-Gurion University of the Negev, 84105, Israel
[2]Department of Geology and Environmental Science, University of Pittsburgh, Pittsburgh, PA15260, USA

**Correspondence:** Liran Goren (gorenl@bgu.ac.il)

**Abstract.** The plan-form geometry of branching drainage networks controls the topography of landscapes as well as their geomorphic, hydrologic, and ecologic functionality. The complexity of networks' geometry shows significant variability, from simple, straight channels that flow along the regional topographic gradient to intricate, tortuous flow patterns. This variability in complexity presents an enigma, as models show that it emerges independently of any heterogeneity in the environmental conditions. We propose to quantify networks' complexity based on the distribution of lengthwise asymmetry between paired flow pathways that diverge from a divide and rejoin at a junction. Using the lengthwise asymmetry definition, we show that the channel concavity index, describing downstream changes in channel slope, has a primary control on the plan-form complexity of natural drainage networks. An analytic model based on geomorphic scaling relations and optimal channel network simulations employing an energy minimization principle reveal that landscapes with low concavity channels attain stable plan-form configuration only through simple geometry. In contrast, landscapes with high-concavity channels achieve plan-form stability with various degrees of network complexity, including extremely complex geometries. Landscape evolution simulations demonstrate that the concavity index and its effect on the multiplicity of available geometries control the tendency of networks to preserve the legacy of former environmental conditions. Consistent with previous findings showing that channel concavity correlates with climate aridity, we find a significant empirical correlation between aridity and network complexity, suggesting a climatic signature embedded in the large-scale plan-form geometry of landscapes.

## 1 Introduction

The plan-form structure of branching fluvial drainage networks has far-reaching implications for the geomorphic, hydrologic, and ecologic functionality of landscapes (Horton, 1945; Sharp and Malin, 1975; Perron et al., 2006; Willett et al., 2018; Pelletier et al., 2018; Stokes and Perron, 2020; Freund et al., 2023; Liu et al., 2024). This structure, which can be expressed based on its geometric and topologic attributes, exhibits significant variation across different regions. In some cases, networks exhibit simple flow paths (Fig. 1a) that generally follow the regional topographic gradient. These flow paths define main drainage basins, draining the main water divide to the mountain front, that are overall similar in shape and size and have a symmetric basin shape with respect to their main trunk (sensu Ramsey et al., 2007). Other networks appear more intricate. These complex networks display tortuous flow paths, asymmetric basin shapes, and varying sizes and shapes of the main basins (Fig. 1b).



Differences in network plan-form complexity directly control the landscape's 3D topography. For the same total relief, longer and more tortuous flow paths have diverse slope aspects and shallower channel slopes, resulting in lower local reliefs (DiBiase et al., 2010) and longer channel segments per elevation range and associated ecoclimatic zone within individual main basins. Conversely, shorter and simpler flow paths that conform to the regional gradient feature a narrower distribution of slope aspects and greater local fluvial relief, such that each main basin is expected to have shorter channel segments within any

given elevation range. These characteristics affect water runoff, sediment transport capacity, rate and pattern of erosion, and the distribution of ecological niches (Rodríguez-Iturbe and Valdés, 1979; Whipple and Tucker, 2002; Badgley et al., 2017; Pelletier et al., 2018; Khosh Bin Ghomash et al., 2019; Beeson et al., 2021; Stokes and Perron, 2020)

    Some of the variability in network complexity could be attributed to the level of heterogeneity in the environmental and boundary conditions affecting the landscape. Spatial gradients in tectonics (Castelltort et al., 2012; Goren et al., 2015, 2014;

Habousha et al., 2023; Cowie et al., 2006; Braun et al., 2013; Mudd et al., 2022), climate (Caylor et al., 2005; Thomas et al., 2011), and lithology (including fabric and fracture density) (Strong et al., 2019; Ward, 2019; Mudd et al., 2022) and discrete geologic structures (Hamawi et al., 2022; Scott and Wohl, 2019) are likely linked to more complex network geometry. However, numerical studies of landscape evolution (Shelef and Hilley, 2014; Tucker and Whipple, 2002; Howard, 1994; Rinaldo et al., 1992; Sun et al., 1994b; Howard, 1990) show that variabilities in complexity emerge even when environmental and boundary

conditions are spatially uniform. This means that drainage complexity could emerge from autogenic network dynamics and be independent of any heterogeneity in the applied forcings.

    The same modeling studies (Shelef and Hilley, 2014; Tucker and Whipple, 2002; Howard, 1994; Sun et al., 1994b; Howard, 1990) found that changing the channel concavity index leads to variations in numerical network complexity, where drainages that are characterized by a higher concavity index are more complex. However, a similar relation was not reported in natural

drainage networks, and the reasoning behind it remained elusive. The concavity index, $\theta$, emerges as the exponent of the globally documented empirical power law relation between the drainage area, $A$, and slope, $S$, known as Flint's law (Flint, 1974; Howard, 1971; Whipple and Tucker, 1999; Willgoose et al., 1991):

$$S = K_s A^{-\theta}, \tag{1}$$

where $K_s$ is referred to as the steepness index. The concavity index describes changes in the inclination of the river channel

along its longitudinal profile as it accumulates drainage area downstream. Empirical studies have found that the concavity index ranges between 0.1-1, with values between 0.3-0.7 being more common (Tucker and Whipple, 2002). A sub-linear profile is characterized by concavity values close to zero, where the channel slope is mostly independent of the drainage area. In contrast, high $\theta$ values, closer to 1, indicate that most of the elevation gain occurs at higher elevations and small drainage areas (Whipple and Tucker, 1999). Channel concavity was shown to vary with channel formative processes (Whipple and Tucker, 1999; Stock

and Dietrich, 2006) and as a reflection of spatial gradients in tectonic uplift (Seybold et al., 2021). Notably, several recent studies identified links between channel concavity and prevailing climatic conditions, particularly aridity. These studies have consistently indicated that arid regions tend to exhibit lower concavity indices (Zaprowski et al., 2005; Chen et al., 2019; Getraer and Maloof, 2021; Michaelides et al., 2022).





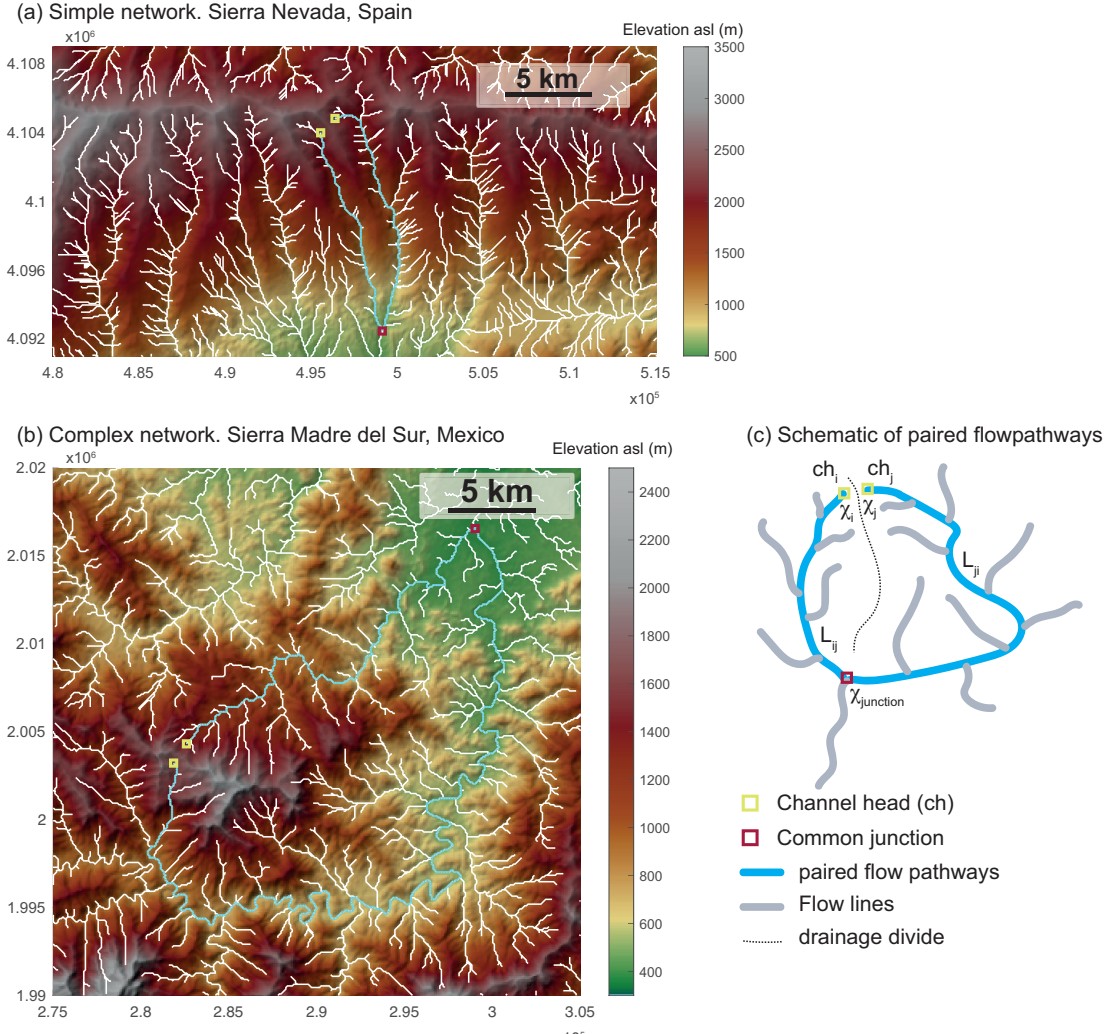

**Figure 1.** Variability of complexity. Examples and schematic representations of paired flow pathways (light blue) originating from two channel heads (yellow squares), diverging from a common divide, and merging at a downstream junction (red square). Hillshade topographies with drainage networks colored in white are displayed in (a) and (b). (a) Shows an example from Sierra Nevada, Spain, where a simple network with sub-parallel trunk streams flows down the main topographic gradient. The median lengthwise asymmetry, $\Delta\mathcal{L}$ over all paired flow pathways in this mountain range is 0.19. (b) Displays an example from Sierra Madre del Sur, Mexico, showcasing a complex drainage network with tortuous flow paths. The median $\Delta\mathcal{L}$ over all paired flow paths in this mountain range is 1.00. (c) A schematic basin with the same components as in panels a and b. $L_{ij}$ and $L_{ji}$, used for calculating $\Delta\mathcal{L}$ of paired flow paths, are measured along the two paired flow paths (light blue) from channel heads, $ch_i$ and $ch_j$ (yellow squares), respectively, to their common junction (red square). $\Delta\chi$ is determined by measuring the $\chi$ values of the channel heads starting at the junction where they merge.





The relationship between channel concavity and network complexity is intriguing because it suggests that the concavity

index, $\theta$, which characterizes the channel longitudinal profiles (Whipple and Tucker, 1999), controls the plan-form properties of entire branching drainage networks. One potential way in which $\theta$ may affect the plan-form geometry of drainage basins is through its hypothesized influence on junction branching angles (Howard, 1971, 1990; Sólyom and Tucker, 2007; Strong and Mudd, 2022). Larger $\theta$ values are associated with larger branching angles, potentially leading to wider basins, while lower $\theta$ values are associated with smaller junction angles and narrower basins. However, consistent changes in the local metric

of junction branching angle, despite their potential effect on basin scaling (Yi et al., 2018), do not necessarily correspond to variations in the multi-scale property of drainage complexity.

An alternative effect relates the concavity index to the lengthwise asymmetry between paired flow pathways that diverge from a single divide and rejoin at a junction or a common base level (Shelef and Hilley, 2014) (Fig. 1c). The degree of lengthwise asymmetry, capturing the multi-scale network geometry, is highly consequential for drainage complexity. Extremely

tortuous and complex networks are linked to large lengthwise asymmetry (Fig. 1b), as one tortuous flow pathway can be meaningfully longer than its across-divide pair. Conversely, simple geometry is linked to small asymmetry (Fig. 1a). Here, we explore the associations between the concavity index and the network complexity as reflected by lengthwise asymmetry. We target observations from natural drainage networks and process-based rationale to quantify and better understand a climate-dependent, first-order control on the 3D geometry of landscapes and the river networks that drain them.

**2  Quantifying the complexity and stability of branching drainage networks**

Drainage divides delineate drainage basins, and consequently, the plan-form geometry of a drainage network is closely tied to the associated drainage divide network (Shelef, 2018; Scherler and Schwanghart, 2020b; Habousha et al., 2023). The stability of the drainage network plan-form geometry is, therefore, tied to the stability of the divide network. When the divides shift or jump, the drainage basin's geometry changes, and the network geometry is not steady. In contrast, as long as the divides

remain stationary, the plan-form geometry of the drainage system is stable, and the network topology is fixed. The stability of drainage divides can be assessed using the gradient of the parameter $\chi$ in between channel heads across divides (Willett et al., 2014), where $\chi$ is proportional to the expected steady-state elevation and is defined as: (Perron and Royden, 2013)

$$\chi(x) = \int_{x_b}^{x} \left( \frac{A_0}{A(x')} \right)^{\theta} dx'. \tag{2}$$

In equation (2), $x$ is the spatial coordinates measured upstream the channel, $x_b$ represents the base level, $x'$ is the integration

parameter, and $A_0$ is a reference drainage area introduced to ensure that the dimensions of $\chi$ [L] are independent of the value of $\theta$.

Considering the $\chi$ values of channel heads (Fig. 1c), when the environmental conditions are spatially uniform, a zero or sufficiently small (Shelef and Goren, 2021) $\Delta\chi$ across divide indicates that the divide is stable (Willett et al., 2014; Shelef and Hilley, 2014), while a large $\Delta\chi$ across a divide could indicate a migrating divide (Willett et al., 2014; Beeson et al., 2017;

Habousha et al., 2023). To compare the stability of divides and drainages of different scales, a normalized $\chi$ difference (referred





to as $\chi$ difference) across divide is defined for paired flow pathways, which originate from two channel heads, $i$ and $j$, across a single divide that join at a downstream junction or base level (Fig. 1c):

$$\Delta\chi_{ij} = \Delta\chi_{ji} = \frac{2|\chi_{ij} - \chi_{ji}|}{\chi_{ij} + \chi_{ji}}. \tag{3}$$

where $\chi_{ij}$ ($\chi_{ji}$) is the $\chi$ value at channel head $i$ ($j$), where $\chi$ is computed between the channel head and the common junction of flow pathways $i$ and $j$.

A formal quantification of network complexity is defined in a similar manner as a measure of normalized lengthwise asymmetry between paired flow pathways (Fig. 1c):

$$\Delta\mathcal{L}_{ij} = \Delta\mathcal{L}_{ji} = \frac{2|L_{ij} - L_{ji}|}{L_{ij} + L_{ji}} \tag{4}$$

where $L_{ij}$ and $L_{ji}$ are the along-flow distances from the two channel heads to their common junction or base level.

The dependency of $\chi$ on $\theta$ and its definition as an integral along flow paths hint at the role of $\theta$ in controlling the complexity of stable plan-form drainage networks. In the extreme case of $\theta = 0$, $\chi_{ij} = L_{ij}$, and $\chi$ difference across divide reduces to lengthwise asymmetry (Shelef and Hilley, 2014). In this case, stable plan-form configurations with $\Delta\chi_{ij} = 0$ for all $i$ and $j$ reduce to $\Delta\mathcal{L}_{ij} = 0$ for all $i$ and $j$. Consequently, for $\theta = 0$, the only stable branching network is one with a perfect lengthwise symmetry, the simplest possible network. As $\theta$ increases, the drainage area distribution along the flow pathways, equation (2), plays a growing role, such that equal $\chi$ across divides could also be achieved when $\Delta\mathcal{L}_{ij} \neq 0$, as long as the drainage area distribution compensates for the lengthwise asymmetry.

## 3    Methods

We explore correlations between landscapes' channel concavity indices and their fluvial branching network complexity while accounting for the networks' stability. The analysis targets natural drainage networks, numerical networks generated using the surface process model DAC (Goren et al., 2014), and numerical optimal channel network simulations (Rinaldo et al., 1992).

### 3.1    Elongated natural mountain ranges

To explore the correlation between channel concavity and drainage network complexity in natural fluvial drainage networks, we independently quantify $\theta$, $\Delta\mathcal{L}$, and $\Delta\chi$ along 18 elongated mountain ranges across the globe. We choose to focus on elongated ranges (rather than study general networks) because (i) such ranges represent topographic units, whose base level boundaries are relatively well-defined, (ii) each of the ranges is relatively simple in terms of its tectonic setting, where the main faults bound the range rather than transect it, and (iii) for each range, $\theta$ and $\Delta\mathcal{L}$ are quantified over a relatively large domain with an along range length between 10s-100s km. We further note that the ranges we choose are situated in both extensional and compressional settings, and accordingly, some are bounded by normal and others by reverse faults. Detailed information about the elongated ranges is listed in Appendix A.

The selection of elongated mountain ranges for the current analysis adhered to specific criteria: (i) The elongated range has a single main divide from which basins drain to two opposite base levels. (ii) There should be a minimum of four basins on



each flank, with the basins' outlets determined by a common elevation contour surrounding the range. (iii) The range should be free from prevalent volcanic characteristics or systematic structural control on the internal drainage pattern.

The analysis of the natural elongated mountain ranges is based on the SRTM 3 arc-seconds Digital Elevation Model (DEM) (Global, 2013) using the TopoToolbox topography analysis package (Schwanghart and Scherler, 2014; Scherler and Schwanghart, 2020a). The boundaries of each range were defined based on a minimum elevation threshold, chosen visually to eliminate alluvial fans and focus on bedrock rivers. Then, interstitial basins were excluded from the analyzed area, such that the analysis was based only on the main basins, draining the main divide to the boundary contour. The drainage network was extracted based on a predefined drainage area threshold, and we explore the sensitivity of the results to the drainage area threshold in
Appendix B.

The concavity index values, $\theta$, for the elongated mountain ranges were determined for the extracted drainage network using the disorder scheme (Gailleton et al., 2021; Hergarten et al., 2016). This method assesses the extent to which the elevation-based order of channel pixels aligns with their order by their $\chi$ value. It involves computing a normalized disorder measure, referred to as $D^*(\theta)$ (Gailleton et al., 2021), for predefined discrete values of $\theta$. The most probable $\theta$ value (referred to as
the best-suited $\theta$) is the one that minimizes the $D^*(\theta)$ metric. The uncertainty in $\theta$ is derived based on the uncertainty in $D^*$ following Gailleton et al. (2021). To evaluate the latter uncertainty, a set of $D^*(\theta)$ values was generated through bootstrapping iterations, with the number of iterations being 1.5 times the number of main basins in each range. In each iteration, $D^*(\theta)$ was computed based on the drainage network of a random selection of 90% of the main basins. The specific parameters used in the bootstrapping iterations were chosen heuristically for their relatively consistent results.

$\Delta\mathcal{L}$ and $\Delta\chi$ were computed for all divide points with distance from divide endpoints exceeding 1000 (Scherler and Schwanghart, 2020a). For each of these divide points, $\Delta\mathcal{L}$ and $\Delta\chi$ were calculated based on the $L$ and $\chi$ values of the two opposing (across the divide point) nearest drainage network pixels along the D8 flow routing raster, and the $L$ and $\chi$ values of the junction (or base level) of the two opposing pathways. To eliminate the influence of overall range asymmetry (which could stem from orographic effects on climate, tectonic advection, or tectonic tilting), divide points from which flow diverges to the two
opposite base levels of the elongated range were excluded from the analysis. The calculation of $\chi$ values employed the most probable $\theta$ value of the range. In cases where the two nearest network pixels were not at the same elevation, a correction was applied to $\Delta\mathcal{L}$ and $\Delta\chi$. This correction adds the values of $A_0^\theta \Delta z_{ij}/K_s$ and $\Delta z_{ij}/K_s A_i^{-\theta}$ to the $\chi$ and $L$ of the lower pixel, respectively. Here, $\Delta z_{ij}$ represents the elevation difference between the two nearest network pixels, $K_s$ is the best-fit steepness index (derived from the slope of the $\chi$-$z$ data of the range), and $A_i$ denotes the drainage area at the lower nearest network pixel
(assuming it is labeled as $i$).

## 3.2 DAC simulations

The DAC landscape evolution model is a processed-based model presented in (Goren et al., 2014). DAC implements an implicit solver of the stream power incision model (Howard, 1994; Whipple and Tucker, 1999): $E = K(PA)^m S^n$, where $E$ [L/T] is erosion rate, $K$ [L$^{(1-3m)}$/T$^{(1-m)}$] is erodibility coefficient, $P$ [L/T] is precipitation rate, $A$ [L$^2$] is drainage area, $S$ [L/L] is
channel gradient, and $m$ and $n$ are positive exponents. Upon identifying $K_s = (E/KP^m)^{1/n}$ and $\theta = m/n$, the stream power



model can be shown to reduce to equation (1). The solver is built upon a triangular, sparse, dynamically adjusting grid. Unlike previous implementations (Goren et al., 2014, 2015; Habousha et al., 2023) that solved for the divide location and identified captures following divide breaching, the DAC implementation used here assumes a strict steepest descent algorithm for flow routing. This choice allows better preservation of the initial conditions (starting with a random subdued topography between 160    0-1 m) and facilitates the comparison of drainage network complexity across concavity values.

For the current analysis, we ran simulations over a domain size of 200 km $\times$ 60 km, producing elongated numerical mountain ranges with a single main water divide. The simulations apply a precipitation rate of $P$=1 m/yr, uplift rate of $U = 0.5$ mm/yr, and a slope exponent, $n$, of 1, such that the concavity index, $\theta$, calculated as $\theta = m/n$, is adjusted by varying the value of the area exponent, $m$. To maintain a consistent global relief despite the changes in $\theta$, the erodibility coefficient $K$ is adjusted 165    across the simulations. Table C1 lists the values of $K$.

The simulations run for 100 million years, ensuring topologic stability by verifying that no alterations in flow routing occurred during the final 10 million years of each simulation. To ensure that the observed effects are indeed related to $\theta$ and not influenced by $K$, Figure C1 replicates a subset of the analysis while keeping $K$ constant.

In the DAC simulations, the calculation $\Delta\mathcal{L}$ and $\Delta\chi$ across divides accounts for all grid-based channel head pairs that share 170    a divide, except for heads located at the base level. When a pair of channel heads does not have the same elevation, a similar correction to the one described for the natural elongated mountain ranges was used with the applied steepness index, $K_s$.

### 3.3 Optimal Channel Network simulations

Optimal Channel Network (OCN) theory (Rodriguez-Iturbe and Rinaldo, 2001; Rinaldo et al., 1992; Molnár and Ramírez, 1998; Banavar et al., 2001) suggests that natural drainage networks self-organize in a way that minimizes global energy expen-175    diture during water flow down the network. In this general view of landscape organization, the energy is evaluated based on the network's topology, represented as sets of nodes and edges, where edges represent the channel connections between nodes. Each internal node has a unique path leading to an outlet node, and the flow paths are loopless. The total energy expenditure, denoted as $P$, for any network defined over the node set is determined by the sum of the local energy expenditures, $P_i$, along each edge, $i$ (Sun et al., 1994b):

$$180 \quad P = \sum_i P_i \propto \sum_i Q_i S_i l_i \propto \sum_i A_i^\gamma l_i = P_{eq}. \tag{5}$$

$S_i$ is the slope across edge $i$, $A_i$ is the upstream drainage area (proxy for discharge $Q$) of node $i$ from which edge $i$ originates, and $l_i$ is the length of edge $i$. The term on the right-hand side of equation (5) is referred to as the energy equivalent, $P_{eq}$. The area exponent, $\gamma$, is expected to correlate inversely with $\theta$, the concavity index (equation 1) (Strong and Mudd, 2022). However, the interdependence of these two exponents as a function of environmental conditions and network hydrology, and their consequent functional relation, remains debated (Strong and Mudd, 2022). Here, we follow the formulation of (Sun et al., 185    1994b) and define $\gamma = 1 - \theta$.

Natural drainage networks were found to resemble numerically generated networks in a state of a local energy minimum (Rodriguez-Iturbe and Rinaldo, 2001; Colaiori et al., 1997). A commonly used criterion to identify and construct such networks





is to ensure that the total energy (i.e., equation 5) is not reduced by a single-edge flip. An edge flip is an operation that redirects an edge that emerges from node $i$ and used to end at node $j$, an immediate neighbor of $i$, to one of its other immediate neighbors, $k \neq j$, without creating loops. A network configuration where any edge flip will only increase its total energy content defines a local energy minimum and corresponds to a topologically stable configuration.

We perform simulations using an iterative greedy algorithm to explore the effect of $\theta$ on networks' evolution toward local energy minimum and the topology of the emerging stable networks. The algorithm starts from a random network configuration, attempts a random edge flip in each iteration, and accepts the new configuration only if the edge flip reduces the total energy. Note that this approach differs from the simulated annealing algorithm (e.g. Sun et al., 1994a, b) that defines a temperature-dependent probability to accept edge flips that increase the total energy as a means to exit local minima and identify a global minimum configuration. Here, we use the greedy approach to eliminate probability-dependent changes that obscure the topological relation between stable networks generated with different $\theta$ values from the same initial random state.

The greedy optimal channel network (OCN) simulations use the computational infrastructure of the TopoToolbox package (Schwanghart and Scherler, 2014). In each iteration, a randomly attempted edge flip is accepted only if it reduces the network's energy. Optimizations with different $\theta$ are initiated from the same random network with a domain size of 200 over 60 nodes. Each node in the domain can drain to one of its eight neighbors, leading edges to be longer in the diagonal than in the rook directions. The nodes along the domain boundary are defined as outlets. Each simulation performed an optimization with a predefined $\theta$ value. The OCN approach lacks a hillslope domain, resulting in a high drainage density. Therefore, to avoid signal overwhelming by channel head pairs that merge at a single node downstream, $\Delta \mathcal{L}$ and $\Delta \chi$ are calculated only for neighboring channel head pairs that drain to different outlets (boundary nodes). Each simulation was run for $3.6 \times 10^6$ iterations.

## 4  Results

### 4.1  Concavity correlates with lengthwise asymmetry in natural and numerical mountain ranges

Among the 18 natural elongated mountain ranges studied, a larger $\theta$ is shown to correlate with a higher median $\Delta \mathcal{L}$ (Fig. 2a blue squares) and a wider spread of $\Delta \mathcal{L}$ values (Fig. 2a blue bars, denoting the 25th and 75th percentiles of the $\Delta \mathcal{L}$ distribution in each range). Though the correlation is not necessarily linear, we quantify it through Pearson's linear correlation. The correlation coefficient between the best-suited $\theta$ and the median $\Delta \mathcal{L}$ is 0.92 (with a slope of 2.16 and P-value of $4.22 \times 10^{-8}$), and the correlation coefficient between the best-suited $\theta$ and the difference between the 75 and 25 percentiles of the $\Delta \mathcal{L}$ distribution is 0.81 (with a slope of 1.30 and P-value of $5.54 \times 10^{-5}$). These trends indicate that those natural networks that are characterized by a higher value of $\theta$ are more complex (with larger median $\Delta \mathcal{L}$) and show greater variability in their level of complexity. In contrast, low $\theta$ natural networks have lower complexity and complexity variation.

To explore the relation between plan-form stability and $\theta$, figure 2a also shows the correlation between $\Delta \chi$ and $\theta$ (green symbols and bars). The Pearson's linear correlation coefficient between the best-suited $\theta$ and the median $\Delta \chi$ is 0.58 (with a slope of 0.33 and P-value of 0.01). The correlation coefficient relating $\theta$ to the difference between the 75 and 25 percentiles of the $\Delta \chi$ distribution is 0.61 (with a slope of 0.43 and P-value of 0.007). Therefore, the association and sensitivity (i.e.,





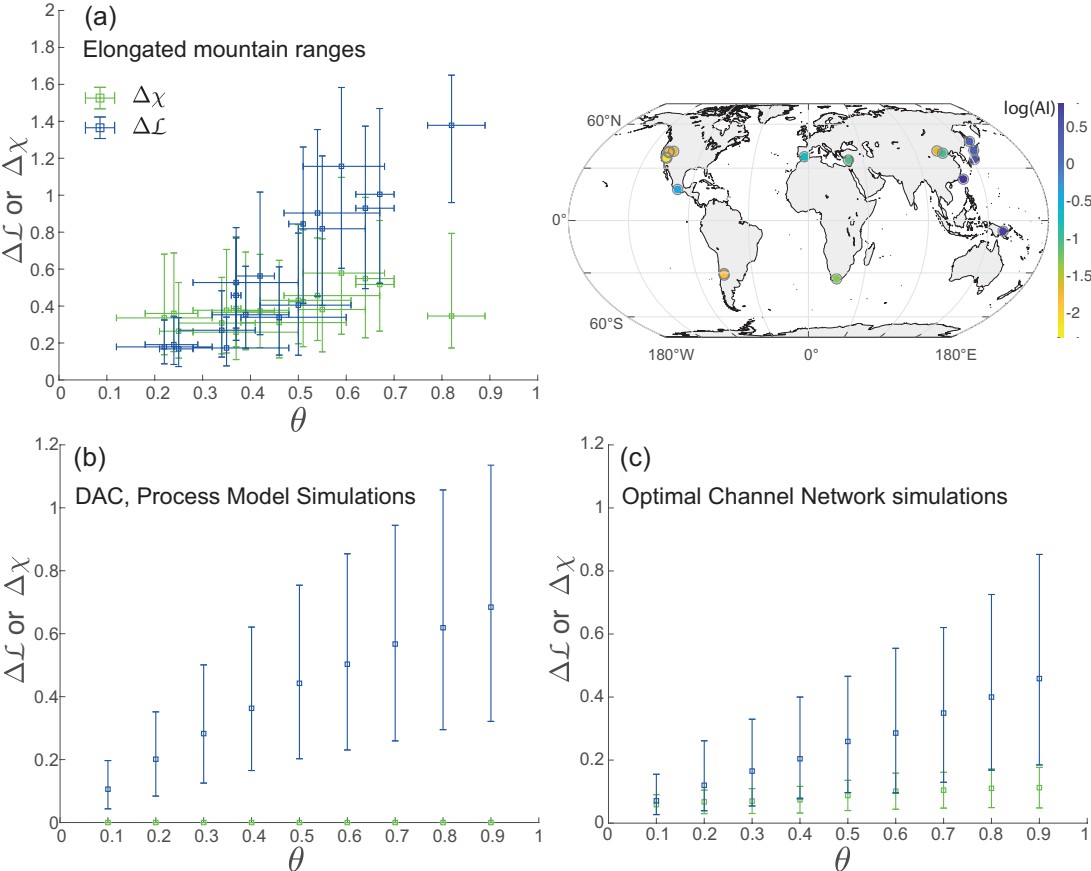

**Figure 2.** Complexity and concavity index. (a) Relations between lengthwise asymmetry, $\Delta\mathcal{L}$ (blue) and $\Delta\chi$ (green), and the concavity index $\theta$. The data represent 18 elongated mountain ranges, visualized as circles on the right-hand side of the map. The circles are color-coded by the logarithm of the Aridity Index (AI) (Zomer et al., 2022), with yellower colors corresponding to more arid conditions. (b) Relations between $\Delta\mathcal{L}$, $\Delta\chi$, and the concavity index $\theta$ for numerical ranges from the DAC process model (Goren et al., 2014). (c) Relations between $\Delta\mathcal{L}$, $\Delta\chi$, and the concavity index $\theta$ for numerical ranges derived from simulations using an optimal channel network model. In all panels, the squares display median values, and the vertical error bars indicate the 25 and 75 percentiles. The horizontal error bars in panel (a) represent the uncertainty in $\theta$, where the square is located at the best-suited $\theta$ value. A relatively high regression slope and significant correlation is observed between $\theta$ and $\Delta\mathcal{L}$, as well as between $\theta$ and the spread of $\Delta\mathcal{L}$ in the natural ranges and simulations. A weaker (a and c) or nonexistent (b) correlation is observed between $\Delta\chi$ and $\theta$.

correlation and slope) between $\Delta\chi$ and its spread to $\theta$ are weaker than those between $\Delta\mathcal{L}$ and its spread and $\theta$. Consequently, the degree of network instability, as quantified by $\Delta\chi$, cannot be invoked as a main driver of the variability in complexity for the analyzed ranges.

225 We cannot fully exclude the possibility that heterogeneity in the environmental condition across the analyzed mountain ranges affects both $\theta$ and $\Delta\mathcal{L}$. To address this possibility, we apply a similar analysis over two types of synthetic, steady-state





landscapes of uniform environmental conditions generated using (i) the DAC process-based landscape evolution model (Goren et al., 2014) and (ii) the process-independent, greedy OCN model. In both models, we run simulations with pre-defined $\theta$ values and measured $\Delta\chi$ and $\Delta\mathcal{L}$ over the emerging drainage networks after they have achieved topologic stability. For $\Delta\mathcal{L}$,

models' results, figures 2b and 2c (blue), show similar trends as those documented for the natural elongated mountain ranges. The median and the spread of $\Delta\mathcal{L}$ increase with increasing $\theta$. For $\Delta\chi$ (blue), model results show constant and infinitesimal values, independent of $\theta$ in the DAC simulations and weak dependency on $\theta$ in the OCN simulations. Notably, while the range of $\Delta\mathcal{L}$ has the same order of magnitude in the natural elongated ranges (Fig. 2a) and the numerical drainages (Fig. 2b&c), the $\Delta\mathcal{L}$ values in the simulations are mostly smaller than the natural $\Delta\mathcal{L}$ for the higher $\theta$ values, potentially revealing the effect of

environmental heterogeneity on the natural terrains complexity, consistent with the higher values of $\Delta\chi$ for these ranges.

## 5 Discussion

### 5.1 Hack's law explains the relation between $\theta$ and $\Delta\mathcal{L}$

To explain the observed correlation between $\theta$ and landscape complexity as quantified by $\Delta\mathcal{L}$, we turn to an analysis of idealized channels. We set two coordinate systems, $x$, that follow paired flow paths from their common junction or base

level to their common divide, such that $x = 0$ is at the junction, and $x = L_i$ ($x = L_j$) is the common divide when measured along channel $i$ ($j$) (Fig. 1c). We assume that the channels obey Hack's law (Rigon et al., 1996), such that the drainage area distribution along the sub-basins whose outlet is the common junction are expressed as follows:

$$A_i(x) = k_{a_i}(L_i - x)^{h_i} \text{ for } 0 \le x \le L_i - x_c = L_{ij} \tag{6}$$

$$A_j(x) = k_{a_j}(L_j - x)^{h_j} \text{ for } 0 \le x \le L_j - x_c = L_{ji}$$

where $k_{a_i}$ and $k_{a_j}$ are Hack's coefficients and $h_i$ and $h_j$ are Hack's exponents. The hillslope length, $x_c$, measured between the divide and the channel heads, is assumed uniform. We further assume that the channels obey a power-low relation between the slope and the drainage area with a concavity index $\theta$, equation (1), and consequently, the $\chi$ values at the channel heads could be defined by combining equations (6) and (2):

$$\chi_i(L_i - x_c) = \int\limits_{x'=0}^{x'=L_i-x_c} \frac{A_0^\theta \mathrm{d}x'}{A_i(x')^\theta} = \begin{cases} \frac{A_0^\theta}{(1-h_i\theta)k_{a_i}^\theta}\left(L_i^{1-h_i\theta} - x_c^{1-h_i\theta}\right) & \text{for } h_i\theta \ne 1 \\ \frac{A_0^\theta}{k_{a_i}^\theta}\ln\left(\frac{L_i}{x_c}\right) & \text{for } h_i\theta = 1 \end{cases} \tag{7}$$

$$\chi_j(L_j - x_c) = \int\limits_{x'=0}^{x'=L_j-x_c} \frac{A_0^\theta \mathrm{d}x'}{A_j(x)^\theta} = \begin{cases} \frac{A_0^\theta}{(1-h_j\theta)k_{a_j}^\theta}\left(L_j^{1-h_j\theta} - x_c^{1-h_j\theta}\right) & \text{for } h_j\theta \ne 1 \\ \frac{A_0^\theta}{k_{a_j}^\theta}\ln\left(\frac{L_j}{x_c}\right) & \text{for} h_j\theta = 1 \end{cases} \tag{8}$$

Requiring the channels to be in a topologic steady-state (stable divide and stable plan-form configuration), the channel heads $\chi$ values across the divide must be equal. For generality and simplicity, we consider the case where $h_i\theta \ne 1$ and $h_j\theta \ne 1$ and write the divide stability criterion, equating $\chi_i(L_i - x_c)$ to $\chi_j(L_j - x_c)$:

$$\frac{1}{(1-h_i\theta)k_{a_i}^\theta}\left(L_i^{1-h_i\theta} - x_c^{1-h_i\theta}\right) = \frac{1}{(1-h_j\theta)k_{a_j}^\theta}\left(L_j^{1-h_j\theta} - x_c^{1-h_j\theta}\right) \tag{9}$$



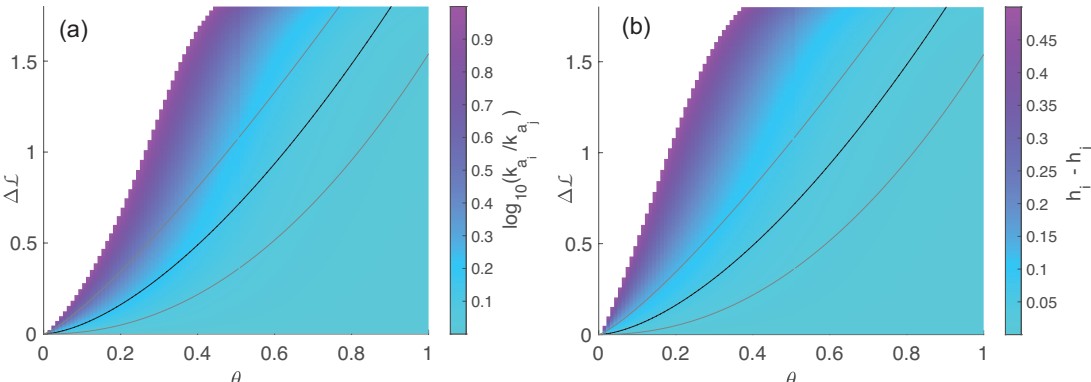

**Figure 3.** Changes in lengthwise asymmetry, $\Delta\mathcal{L}$ as a function of concavity index, $\theta$ with differing Hack's law parameters, equation (6). $\Delta\mathcal{L}$ is calculated for idealized sub-basins along paired flow pathways, $i$ and $j$, sharing a stable divide (Fig. 1c). (a) The color map represents the logarithm of Hack's coefficients' ratio, assuming that Hack's exponents are identical, $h_i = h_j = 2$. (b) The color map represents the difference between Hack's exponents, assuming Hack's coefficients are identical and $h_i = 2$. In both panels $L_i = 50$ km, $L_j \leq L_i$, and $x_c = 0.25$ km. The black and gray curves represent fitted power laws to the $\Delta\mathcal{L}$-$\theta$ trends based on the natural elongated mountain ranges shown in figure 2a. For the median $\Delta\mathcal{L}$ the fit is $\Delta\mathcal{L} = 2.12\theta^{1.60}$ (black curves); for the 75 percentile the fit is $\Delta\mathcal{L} = 2.49\theta^{1.23}$ (upper gray curves); for the 25 percentile the fit is $\Delta\mathcal{L} = 1.54\theta^{2.14}$ (lower gray curves).

If Hack's coefficients and exponents are identical for the two sub-basins, i.e., $k_{a_i} = k_{a_j}$ and $h_i = h_j$ then the only solution to equation (9), is lengthwise symmetry, $L_i = L_j$ and $\Delta\mathcal{L} = 0$ for all values of $\theta$.

To explore the possibility of stable topologic configurations that respect equal $\chi$ across divide while permitting $\Delta\mathcal{L} \neq 0$, we relax the restricting assumption of identical Hack's coefficient or exponent. First, we consider the case where $k_{a_i}/k_{a_j}$ is not necessarily 1 whereas $h_i = h_j = h$. Figure 3a shows the value of $k_{a_i}/k_{a_j}$ needed to ensure equal $\chi$ across divide, equation
(9), as a function of $\theta$ and $\Delta\mathcal{L}$. Without loss of generality, we assume that $L_j < L_i$, and consider values of $k_{a_i}/k_{a_j} < 10$, approximately a factor of three larger than the range of reported natural values (Montgomery and Dietrich, 1992; Mueller, 1972; Willemin, 2000; Dodds and Rothman, 2000; Shen et al., 2017; Sassolas-Serrayet et al., 2018). Considering a fixed $\Delta\mathcal{L}$, the figure shows that for low values of $\theta$, there are no $k_{a_i}/k_{a_j}$ values within the range for which $L_i \neq L_j$. As $\theta$ increases, $\Delta\mathcal{L} \neq 0$ could be achieved with high $k_{a_i}/k_{a_j}$ values. As $\theta$ further increases, the $k_{a_i}/k_{a_j}$ values ensuring stable configurations
with any particular $\Delta\mathcal{L}$ become smaller. This analysis reveals that for small $\theta$ values, stable topologies can be achieved only with small $\Delta\mathcal{L}$, but as $\theta$ increases, small differences in Hack's coefficients permit stable topological configuration with large lengthwise asymmetry.

Next, we consider the case where $h_i \neq h_j$ whereas $k_{a_i} = k_{a_j} = k_a$. Here, equation (9) becomes independent of $k_a$, and for a fixed value of $h_i$, the value of $h_j$ for different $\Delta\mathcal{L}$ can be solved only implicitly. Figure 3b shows the value of the difference
$h_i - h_j$ as a function of $\theta$ and $\Delta\mathcal{L}$, for $h_i = 2$. The difference is used rather than the ratio, as with $k_a$ (Fig. 3a), because the $h$ exponents vary by a factor, whereas the $k_a$ coefficients can vary by orders of magnitude. We only consider solutions where $0 < h_i - h_j < 0.5$, again representing a difference larger by a factor of approximately three with respect to the range of $h$





exponents reported for natural terrains. Here, as well, for low values of $\theta$, a solution exists only for very small $\Delta\mathcal{L}$, and as $\theta$ increases, smaller differences in the Hack's exponents allow for a stable configuration with a large lengthwise asymmetry.

This analysis reveals that the widely documented geomorphic relationships of Hack's law, equation (6), and Flint's law, equation (1), are consistent with and can explain natural and numerical observations (Fig. 2) of the relations between $\theta$ and $\Delta\mathcal{L}$. More specifically, the analysis shows that when $\theta$ is large, stable configurations with zero $\Delta\chi$ across divides can be achieved even when $\Delta\mathcal{L} \gg 0$, by exploiting small variations in Hack's exponent and coefficient. When $\theta$ is small, topological stability necessitates high lengthwise symmetry (small $\Delta\mathcal{L}$).

The curves that overly figure 3 show the best-fit power law relation between the best-suited $\theta$ and the median (black curve) and 25 and 75 (gray curves) percentile of $\Delta\mathcal{L}$ based on the elongated mountain ranges shown in figure 2a. The relation between the curves and the colormap that underlies them reveals that high $\theta$ natural networks achieve large $\Delta\mathcal{L}$ by exploiting small variations in Hack's exponent or coefficient. In contrast, low $\theta$ networks require greater variability in Hack's parameters to achieve a much smaller $\Delta\mathcal{L}$.

## 285  5.2   Optimal Channel Networks (OCN) perspective

Whereas Hack's law-based analysis, section 5.1, explains the observed correlation between concavity and complexity from a geomorphic scaling relations standpoint, the OCN framework helps conceptualize this correlation based on energy considerations. Figure 4a shows the evolution of the normalized energy equivalent, $P_{eq}/P_{eq_{init}}$ as a function of iteration number during the energy optimization process under different values of $\theta$ and starting from the same initial random network, with equivalent 290  energy $P_{eq_{init}}$. The figure shows that as $\theta$ decreases, the normalized energy equivalent reduction is greater. Figure 4b shows the evolution of the median $\Delta\mathcal{L}$ during the optimization procedure, displaying a similar trend to that of the normalized energy equivalent, with a greater reduction in $\Delta\mathcal{L}$ with decreasing $\theta$. Notably, while the greedy algorithm ensures a monotonous energy reduction with an increasing number of iterations (Fig. 4a), the $\Delta\mathcal{L}$ trends are non-monotonous (Fig. 4b). The $\Delta\mathcal{L}$ values of the final energy minimum networks are depicted in figure 2c. Appendix D discusses the trend of $\Delta\chi$ through the 295  optimization iterations.

     Examples of network topology optimized using the greedy algorithm with different values of $\theta$ are shown in figure 5. High $\theta$ values result in complex, tortuous networks that do not significantly differ from the random initial conditions, consistent with the low number of accepted edge flip operations (Fig. 4c). As $\theta$ decreases, the networks become less complex, and the legacy of the initial conditions is gradually erased (Figs. 5c and d). A similar behavior was recorded in surface process model 300  simulations (Shelef and Hilley, 2014; Kwang and Parker, 2019; Howard, 1994).

     The reduction in the minimum normalized energy equivalent and network complexity observed in figures 4a and 4b, and the gradual deviation from the random initial network with decreasing $\theta$ (seen in figure 5) could be rationalized analytically. In the limit of $\theta = 0$ and when the edge length is uniform, the energy equivalent, equation (5), $\sum_i A_i^{1-\theta} l_i$, becomes proportional to $\sum_i L_i$ (here, $l_i$ is the length of a single edge and $L_i$ represents the distance to the outlet node) (Colaiori et al., 1997). In this case, 305  the global minimum is attained when each node drains to an outlet along the shortest path, contributing its local area to the minimal number of nodes. This ensures that in the limit of $\theta \to 0$, the emerging topology is such that each $L_i$ is minimal and



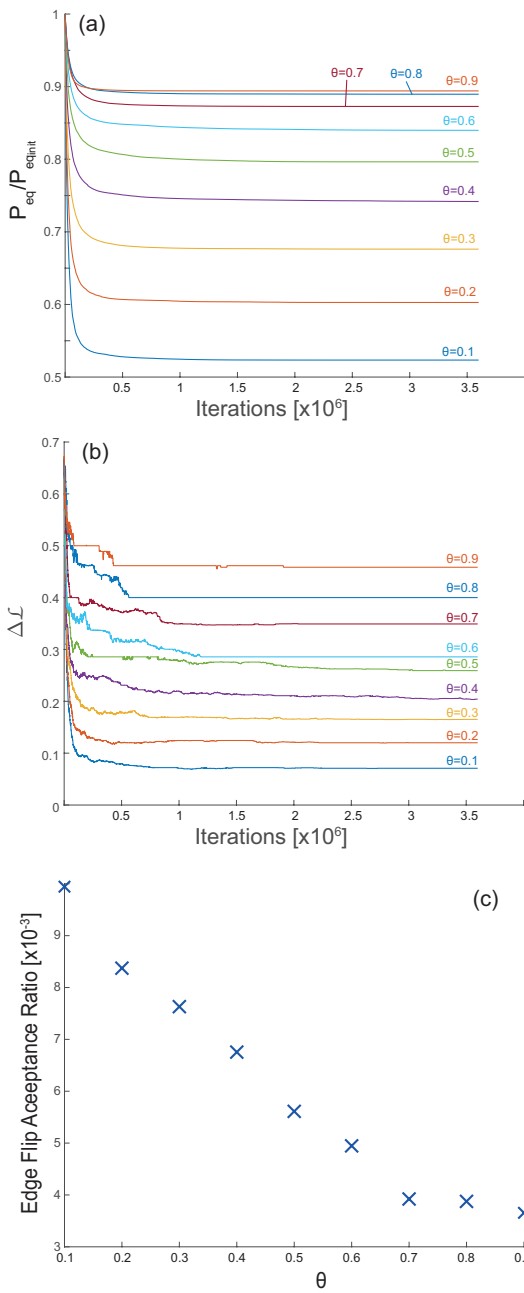

**Figure 4.** Optimal Channel Network (OCN) dynamics as a function of concavity index. Transient response and steady-state values from OCN simulations (see Method section for details) showing the effect of the concavity index, $\theta$ on the reduction trends of the (a) Normalized energy equivalent, $P_{eq}/P_{eq_{init}}$, equation (5), (b) Median $\Delta\mathcal{L}$, and (c) The acceptance ratio of edge flip operations that reduce the total energy. I.e., the quotient of edge flips and the total number of iterations.





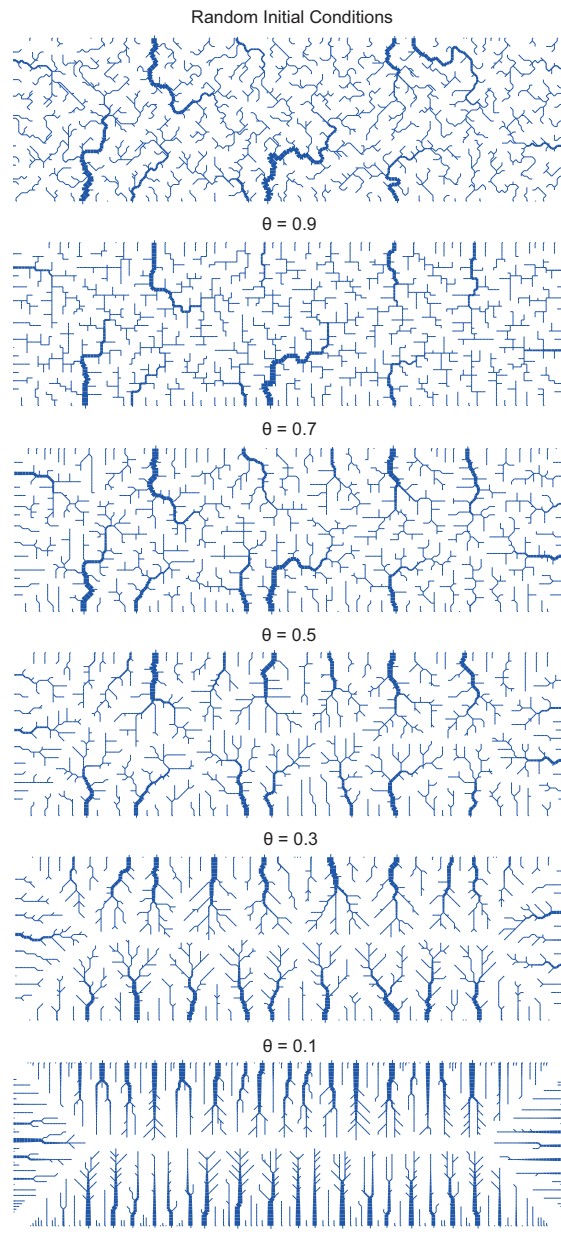

**Figure 5.** Topologies emerging from OCN simulations with different concavity index. Random initial conditions (top) and final, steady optimal channel networks following the application of the greedy algorithm with different $\theta$ values. Note the great complexity and the similarity to the initial conditions of the high $\theta$ networks relative to the simple geometry of the low $\theta$ networks that significantly differ from the initial conditions. Edges are plotted only for drainage area > 5 nodes, and edge width scales with the number of draining nodes.

therefore equal across all divides and $\Delta\mathcal{L} \to 0$. Such an optimal network exhibits an exceptionally simple geometry that differs significantly from any random network, explaining the low-complexity configurations that characterize low $\theta$ OCNs (Fig. 5),



the many edge flips needed to achieve such configurations, and the associated large normalized energy equivalent reduction
(Figs. 4a and 2c).

In the $\theta \to 1$ limit, the energy equivalent becomes independent of the drainage area. Assuming uniform $l_i$, the energy equivalent is a function of the number of nodes in the domain and is independent of the specific drainage configuration. Consequently, all networks, including any random initial complex network, have the same minimal energy. This, in turn, explains the low number of edge flips when $\theta \to 1$, the small reduction in normalized energy equivalent, and the similarity of
315 the final optimal OCN to the initial random network.

For $0 < \theta < 1$, our analysis reveals monotonous relation between $\theta$ to the normalized energy equivalent reduction (Fig. 4a), the final $\Delta\mathcal{L}$ (Fig. 2c) and the acceptance ratio of edge flips (Fig. 4c). Overall, the OCN analysis indicates that $\theta$ determines the multiplicity of stable topologies. Networks with a high $\theta$ value can achieve stability across a wide range of $\Delta\mathcal{L}$ values, allowing for many possible stable topologies. This includes the formation of complex networks with large $\Delta\mathcal{L}$ values, similar
to random networks. In contrast, to attain stability, networks with a low $\theta$ value are restricted to simple topologies with smaller $\Delta\mathcal{L}$ values.

### 5.3 Climate aridity controls network complexity

Large data compilations of river profiles revealed that channel concavity correlates with climatic and hydrologic factors. More specifically, mean annual rainfall and rainfall intensities correlate positively with the concavity index (Zaprowski et al., 2005).
Likewise, the degree of aridity as quantified by the aridity index, the quotient of precipitation and evapotranspiration potential (Zomer et al., 2022), correlates with channel concavity, such that in arid regions (with low aridity index) rivers are less concave (Chen et al., 2019; Getraer and Maloof, 2021). Combining these established relations between climate and concavity index and the correlation identified here between the concavity index and drainage complexity implies that the climatic conditions at which drainage networks develop could be encoded in their complexity and, thus, in the large-scale plan-form geometry
of landscapes. Consequently, arid climates, characterized by low channel profile concavity, likely favor the development of low-complexity networks, whereas a more humid climate, characterized by high-conavity channels, is expected to result in variable complexity, including high-complexity drainage networks.

We examine the relationship between the complexity, $\Delta\mathcal{L}$, and the aridity index across the elongated natural mountain ranges. The Aridity Index for each elongated range is calculated based on pixel statistics of the Global Aridity Index raster
(Zomer et al., 2022). A mask based on the analyzed area in each elongated range is used to extract the relevant pixels from the AI raster. The median and 25 and 75 percentile of the pixel values within each such mask are used in the analysis.

Figure 6 shows a positive correlation between climate aridity index (AI) and network complexity for the elongated mountain ranges. The Spearman's rank correlation coefficient for this correlation is 0.59 with a P-value of 0.01, indicating that higher complexity tends to be associated with higher AI, representing more humid climates. It's worth noting the significance of this
correlation, considering that our dataset comprises only 18 mountain ranges. This finding is surprising because previous studies that explored the link between aridity and concavity relied on much larger datasets, often including several hundred thousand data points, to identify a signal out of the background noise (Chen et al., 2019; Getraer and Maloof, 2021). This could be seen





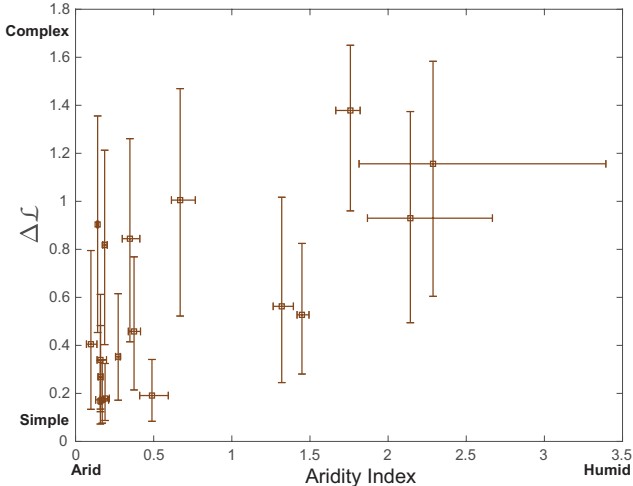

**Figure 6.** Complexity and Aridity. The relation between drainage complexity, $\Delta\mathcal{L}$, and Aridity Index, the quotient of precipitation and evapotranspiration potential (Zomer et al., 2022), for the 18 elongated mountain ranges analyzed in figure 2a. The box symbols represent the median values, and the bars show the 25 and 75 percentiles. A significant correlation (P = 0.01) with a Spearman rank correlation coefficient of 0.59 indicates that more complex networks (high $\Delta\mathcal{L}$) are associated with more humid climatic conditions and their corresponding hydrology.

as another support for the strong link between the network plan-form complexity and the formative concavity index (equation 1), which is expected to strongly depend on the hydrologic conditions (Whipple and Tucker, 1999; Freund et al., 2023).

### 5.4 Concavity controls plan-form landscape evolution

The results so far reveal that high-concavity landscapes can achieve topologic stability with variable complexity, whereas the stability of low-concavity landscapes is conditioned by low complexity. These findings are consequential for landscape evolution, which we further investigate and quantify using landscape evolution simulations in DAC.

#### 5.4.1 Changing climate

A first simulation set is designed to examine how the drainage network adjusts to changes in the concavity index, reflecting changing climatic, hydrologic, and geomorphic conditions. Previous studies have linked the concavity index to channel-forming processes, where debris flow channels typically exhibit lower concavity compared to fluvial channels (Stock and Dietrich, 2006). Therefore, a decrease in the concavity index can represent aridification or a transition to a debris-dominated landscape, while an increase in concavity may indicate a transition to a more humid climate or a fluvial-dominated regime.

The simulation set starts with a topologically stable landscape of high concavity ($\theta = 0.9$) and high complexity (brown frame in figure 7). We then gradually decrease the concavity index by increments of 0.1. Following each increment, we let the landscape re-equilibrate until no further topological changes are observed. We measure the median value of $\Delta\mathcal{L}$ for this



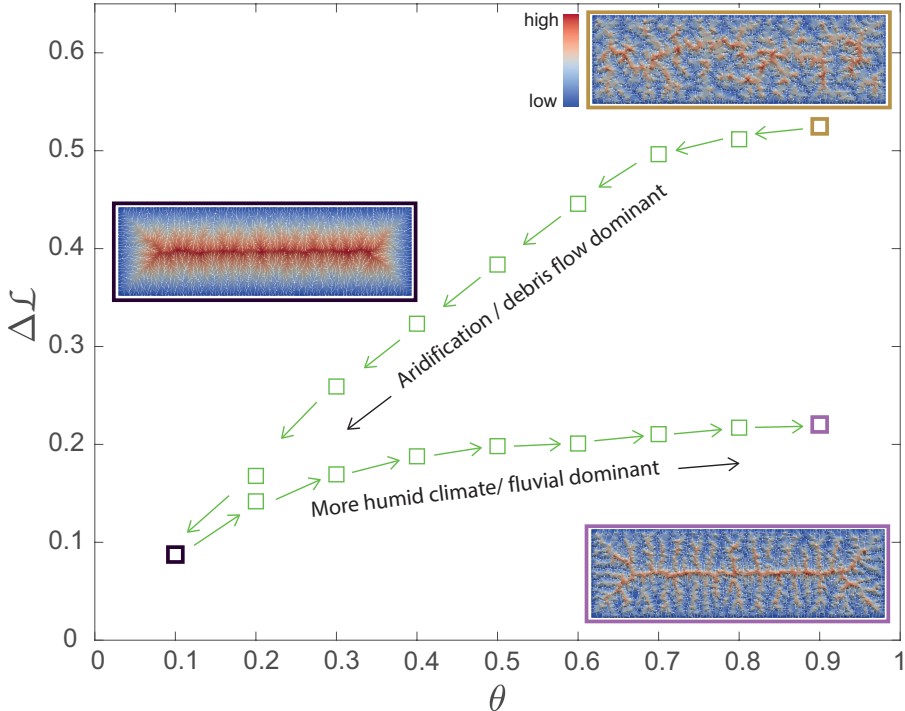

**Figure 7.** The effect of changing concavity on network complexity. Simulation results from the DAC process model (Goren et al., 2014) showing the relation between the complexity, $\Delta\mathcal{L}$ and concavity index, $\theta$ during a two-stage scenario. First, the concavity is gradually decreased by steps of 0.1 (left pointing arrow), and second, the concavity increases in steps of 0.1 (right-pointing arrows). The network and topography of the initial, $\theta = 0.9$ conditions are depicted in the brown framed topography, the $\theta = 0.1$ landscapes, corresponding to the end of the first stage and beginning of the second stage is shown with a black framed topography, and the final $\theta = 0.9$ landscape is shown with a purple framed topography. Note (i) the hysteresis response of $\Delta\mathcal{L}$, showing different trends depending on the directional change in the concavity index, and (ii) the difference in topographic complexity between the two $\theta = 0.9$ maps. For the current analysis, $\Delta\mathcal{L}$ is calculated only for channel heads that drain to different outlets, such that the shared junction is the base level. Measuring $\Delta\mathcal{L}$ over this longer length scale emphasizes the hysteresis signal.

equilibrated landscape and then use this landscape as the initial condition for the next increment. The procedure continues until reaching a low concavity value of $\theta = 0.1$. Subsequently, we gradually increase the concavity index by 0.1, following the same re-equilibration procedure, until returning to the initial concavity value of $\theta = 0.9$.

Figure 7 shows that during the decreasing concavity stage, the median $\Delta\mathcal{L}$ gradually decreases, consistent with the results shown in figure 2b. However, in the increasing concavity stage, a hysteresis response is observed. The median $\Delta\mathcal{L}$ in the increasing concavity stage is lower than that of the same concavity value in the decreasing concavity stage and overall shows only a slight increase compared to the median $\Delta\mathcal{L}$ of the landscape with $\theta = 0.1$ (black frame in figure 7). Consequently, when the concavity index returns to its initial value of $\theta = 0.9$ (purple frame in figure 7), the median $\Delta\mathcal{L}$ of the landscape is smaller by a factor of 2.4 with respect to the median $\Delta\mathcal{L}$ of the initial conditions with the same concavity index.





The dynamics depicted in figure 7 suggest that when aridification or a transition to a debris flow-dominated regime takes place, there is a significant autogenic reorganization of the drainage network towards a lower complexity configuration. In contrast, when transitioning to a more humid climate or a fluvially dominated regime, minimal reorganization is expected, and the resulting landscape may retain the complexity of the antecedent, more arid or debris-flow-dominated state. This implies that the complexity of a given landscape, as reflected by $\Delta\mathcal{L}$, is influenced by the lowermost concavity experienced by this landscape. For example, a low complexity landscape, currently located in a humid climate (and exhibiting high concavity), might suggest a formation or modification history under drier (i.e., low concavity) conditions. Differences in past aridity can, therefore, be invoked to explain the variability in $\Delta\mathcal{L}$ with aridity (Fig. 6), as well as the increased variability in $\Delta\mathcal{L}$ with concavity in the elongated mountain ranges (Fig. 2a).

### 5.4.2 Fingerprints of antecedent lithologic conditions

To explore another scenario in which the complexity records the legacy of past conditions, we focus on the effect of lithology in a second set of simulations. Here, drainage networks evolve from a subdued, random topography, similar to the simulations depicted in Figure 2b. However, in this case, a narrow, one-kilometer-wide slab protrudes into the surrounding rocks. The slab extends down to a depth of five kilometers into the crust and is positioned midway between the center of the domain and the southern base level of the evolving range, as shown in Figure 8. The slab's erodibility is higher by a factor of 100 compared to that of the surrounding rocks. This high erodibility slab can be conceptualized as a fault zone containing crushed, more erodible rocks.

During the initial period of approximately 10 Myr (with an imposed uplift rate of $5\times10^{-4}$ m/yr) the developing channel networks incise into the layers of rock protruded by the higher erodibility slab. Once the slab is fully removed by erosional exhumation of five kilometers, the networks continue to incise into rocks with a uniform erodibility.

The simulation results reveal distinct behaviors of the drainage networks over time, depending on the concavity index and the presence of a high-erodibility slab. In the first 10 Myr, south-draining channel segments favor the high-erodibility slab, resulting in channel segments parallel to the southern edge of the mountain range (Duvall et al., 2020). However, beyond this period, after the high-erodibility slab has been eroded, the response of the drainage networks diverges based on the concavity index.

To quantitatively assess this response, we analyze the number of longitudinal channel segments whose orientation is within 10 degrees of the slab's orientation, where a segment connects two neighboring numerical nodes. Figure 8 illustrates the ratio of longitudinal segments after 100 Myr (post-slab removal by erosion) to the number of longitudinal segments at 10 Myr, prior to the complete exhumation of the slab, for south draining basins. As the concavity index increases, a greater preservation of horizontal segments is observed, although the slab is completely eroded. In the case of a landscape with $\theta = 0.9$ (right-hand side of Figure 8), the drainage pattern predominantly retains the legacy of the high-erodibility slab, resulting in a preservation ratio of longitudinal segments close to 1. Conversely, a landscape with $\theta = 0.2$ (left-hand side of Figure 8) is characterized by lower preservation of longitudinal segments after the removal of the high-erodibility slab, leading to a low preservation ratio.



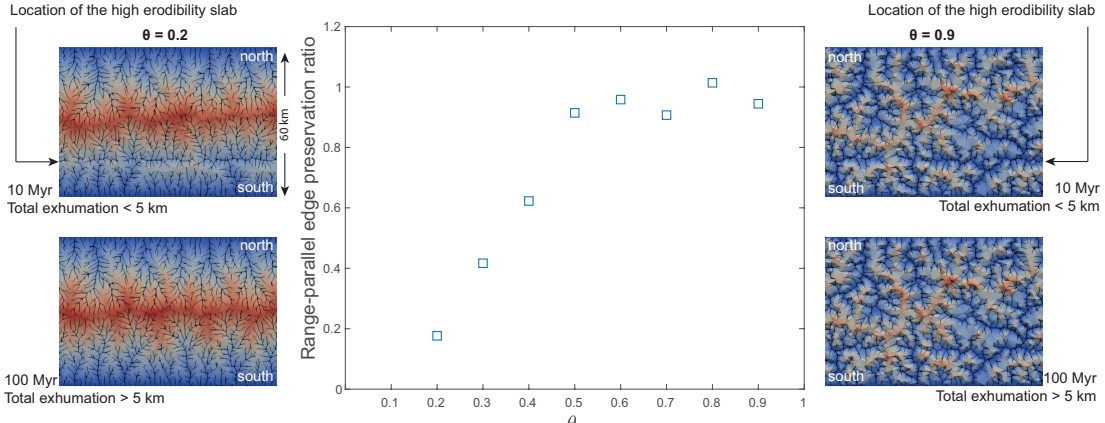

**Figure 8.** Preservation of antecedent conditions as a function of concavity index in a DAC (Goren et al., 2014) process model simulation series. In these simulations, drainage networks evolve primarily through the incision of spatially uniform rocks, except for a narrow, elongated slab with a higher erodibility (100 times greater), a width of one kilometer, and a depth that extends from the surface to 5 km into the crust. The slab is located 15 kilometers from the southern base level. The south-draining basins originally incise into the slab-containing rocks, and the drainage network locally aligns with the high erodibility slab, forming longitudinal channels regardless of the concavity index (topographies at the top of the figure). Following a 5-kilometer exhumation, the slab is fully eroded away, and rivers incise into uniformly erodible strata. Drainage network response to the removal of the slab depends on the concavity index. In simulations of low concavity (topographic maps on the left), significant drainage reorganization removes the longitudinal segments (topography at the bottom left). In contrast, high concavity landscapes (topographic maps to the right) preserve the antecedent longitudinal segments after the slab is eroded (topography at the bottom right). The central graph presents the preservation ratio by counting the number of longitudinal segments at 100 Myr (after slab removal by erosion) and dividing it by the number of longitudinal segments at 10 Myr (while the high-erodibility slab is still present) for all south draining basins.

The slab simulation set reveals that environmental conditions promoting sustained high concavity over long timescales result in drainage networks that record the cumulative effect of spatially varying heterogeneities (e.g., lithology) even long after these heterogeneities were removed. In contrast, environmental conditions associated with low concavities might eliminate traces of past spatial heterogeneities.

The two simulation sets indicate that landscapes with high concavity indexes are less sensitive to variations in exogenic
forcing. As a result, high concavity index landscapes can preserve the original drainage patterns (Kwang and Parker, 2019), their specific complexity, and the legacy of the environmental conditions in which these landscapes initially formed. Conversely, networks with low concavity indexes undergo reorganization to attain a configuration of lower complexity in response to any environmental, climatic, and tectonic changes. Consequently, they possess a limited capacity to retain the legacy of the previous environmental conditions.





## 6   Conclusions

The current analysis reveals that channel concavity index, $\theta$, reflecting the channel longitudinal profile, sets a first-order control on the plan-form complexity of drainage networks as quantified by the statistics of asymmetry in the length of paired flow pathways, $\Delta\mathcal{L}$. Variability of concavity indices, thus, explains the observed variability in complexity across the globe. Specifically, $\theta$ controls the multiplicity of stable plan-form configurations available to a drainage network. When $\theta$ is small, the number of stable configurations is small, and they are all characterized by high lengthwise symmetry, producing simple-looking drainage networks. When $\theta$ is large, the number of stable plan-form configurations increases, and they include configurations with a large degree of lengthwise asymmetry, producing complex geometry. Consequently, high $\theta$ drainages can be found in topological stable configurations that are characterized by high lengthwise asymmetry (i.e., high complexity), whereas the stability of low $\theta$ drainages is conditioned by a smaller lengthwise asymmetry (i.e., low complexity). These findings can be theoretically explained based on an energy minimization principle or by combining two empirical power laws that are readily documented across the globe: Hack's law, equation (6), and Flint's law, equation (1), describing the relation between drainage area to channel length and channel slope, respectively. The multiplicity of steady configurations of high $\theta$ landscapes further means that the planform geometry of these networks more readily preserves the legacy of former conditions (Kwang and Parker, 2019).

Drainage network complexity of elongated mountain ranges correlates with the aridity index, a measure of climate dryness. The correlation emerges despite the relatively small number of natural ranges we analyzed and is intuit through the effect of climate on channel formative concavity (Whipple and Tucker, 1999). Therefore, the geometric complexity of drainage networks over entire mountain ranges records information about prevailing climatic conditions.

*Code availability.*   The code used for the (i) analysis of the elongated mountain range, (ii) production and analysis of the DAC numerical landscapes, (iii) production and analysis of the greedy OCN landscapes, and (iv) the Hack's law based analysis is available at https://github.com/gorenl/ConcavityControlsComplexity.

*Data availability.*   Appendix A and Table A1 lists information about the 18 elongated mountain ranges analyzed here

.

## Appendix A:   Elongated mountain ranges

Comprehensive details of the 18 elongated mountain ranges and the data utilized for generating figures 2a and 6 is listed in Table A1. The ranges' relief and drainage network are depicted in figure A1.




**Table A1.** Elongated Mountain Ranges[1]

| Range Name | Lat[2] | Lon[2] | $z_{max}$ [m][2] | $z_{th}$ [m][3] | A[km$^2$][4] | $\theta$ | $K_s$[5] | $\Delta\mathcal{L}$ | $\Delta\chi$ | AI[6] | Pairs[7] |
|---|---|---|---|---|---|---|---|---|---|---|---|
| Central Mountain Range, Taiwan[8] | 23.47 | 120.96 | 3917 | 80 | 11991 | 0.59 (0.51,0.68) | 0.56 | 1.16 (0.60,1.58) | 0.58 (0.25,1.10) | 2.29 (1.81,3.39) | 57010 |
| Clan Alpine Mountains, Nevada | 39.69 | -117.87 | 2677 | 1700 | 204 | 0.34 (0.25,0.41) | 0.15 | 0.27 (0.12,0.48) | 0.31 (0.14,0.56) | 0.16 (0.14,0.17) | 1547 |
| Daqing Shan, China | 40.71 | 109.12 | 2293 | 1200 | 297 | 0.55 (0.50,0.64) | 0.17 | 0.82 (0.40,1.21) | 0.38 (0.15,0.76) | 0.19 (0.17,0.20) | 1961 |
| Finisterre Range, Papua New Guinea[8] | -5.95 | 146.38 | 4096 | 400 | 8227 | 0.64 (0.62,0.70) | 0.99 | 0.93 (0.49,1.37) | 0.55 (0.23,0.99) | 2.14 (1.87,2.67) | 31675 |
| Humboldt Range, Nevada | 40.52 | -118.17 | 2984 | 1450 | 291 | 0.46 (0.38,0.60) | 0.21 | 0.34 (0.13,0.61) | 0.31 (0.12,0.65) | 0.16 (0.14,0.20) | 2313 |
| Inyo Mountains, California | 36.71 | -117.96 | 3363 | 1200 | 266 | 0.35 (0.28,0.48) | 0.37 | 0.17 (0.08,0.34) | 0.38 (0.15,0.71) | 0.17 (0.13,0.21) | 2419 |
| Kammanassie Mountains, South Africa | -33.62 | 22.94 | 1935 | 630 | 353 | 0.39 (0.32,0.48) | 0.16 | 0.35 (0.17,0.62) | 0.37 (0.16,0.69) | 0.27 (0.26,0.29) | 2858 |
| Lüliang Mountains, China | 39.27 | 112.96 | 2391 | 1100 | 1036 | 0.37 (0.36,0.38) | 0.14 | 0.46 (0.21,0.77) | 0.39 (0.18,0.78) | 0.37 (0.34,0.42) | 5685 |
| Panamint Range, California | 36.17 | -117.09 | 3344 | 800 | 713 | 0.50 (0.42,0.61) | 0.40 | 0.40 (0.13,0.80) | 0.43 (0.20,0.84) | 0.10 (0.07,0.14) | 6852 |
| Sakhalin Mountains, Russia | 47.07 | 142.88 | 1028 | 60 | 556 | 0.42 (0.37,0.45) | 0.09 | 0.56 (0.25,1.02) | 0.37 (0.18,0.68) | 1.32 (1.26,1.39) | 5774 |
| Sierra del Valle Fértil, Argentina | -30.44 | -67.80 | 2311 | 1050 | 696 | 0.54 (0.47,0.67) | 0.20 | 0.90 (0.45,1.36) | 0.46 (0.21,0.77) | 0.14 (0.14,0.15) | 6415 |
| Sierra Madre del Sur, Mexico[8] | 17.52 | -100.31 | 3105 | 380 | 8406 | 0.67 (0.62,0.70) | 0.63 | 1.01 (0.52,1.47) | 0.52 (0.26,0.86) | 0.67 (0.61,0.77) | 17271 |
| Sierra Nevada, Spain | 37.05 | -3.31 | 3446 | 1200 | 628 | 0.24 (0.18,0.29) | 0.18 | 0.19 (0.08,0.34) | 0.36 (0.15,0.69) | 0.49 (0.41,0.59) | 6046 |
| Sierra Pie de Palo, Argentina | -31.32 | -67.92 | 3157 | 650 | 882 | 0.25 (0.21,0.28) | 0.17 | 0.17 (0.07,0.34) | 0.26 (0.12,0.53) | 0.16 (0.15,0.17) | 9731 |
| Toano Range, Nevada | 40.50 | -114.30 | 2914 | 1710 | 364 | 0.22 (0.12,0.32) | 0.07 | 0.18 (0.09,0.32) | 0.34 (0.14,0.68) | 0.19 (0.16,0.22) | 3721 |
| Troodos Mountains, Cyprus | 34.94 | 32.86 | 1949 | 200 | 1899 | 0.51 (0.48,0.51) | 0.18 | 0.84 (0.42,1.26) | 0.42 (0.18,0.81) | 0.35 (0.30,0.41) | 18104 |
| Tsugaru peninsula, Japan | 40.97 | 140.55 | 666 | 30 | 263 | 0.37 (0.28,0.50) | 0.05 | 0.53 (0.28,0.82) | 0.26 (0.11,0.52) | 1.45 (1.42,1.49) | 2343 |
| Yoro Mountains, Japan | 35.28 | 136.51 | 885 | 130 | 84 | 0.82 (0.77,0.89) | 0.20 | 1.38 (0.96,1.65) | 0.35 (0.17,0.79) | 1.76 (1.67,1.82) | 726 |

[1] Data based on threshold drainage area of 145 pixels corresponding to $\approx 1$ km$^2$. Values to the left of the parenthesis are the median. Values in parentheses correspond to the 25th and 75th percentiles of the population.

[2] Latitude, Longitude, and elevation of the highest peak in the range.

[3] Elevation of the minimal closed contour defining the bounds of the range.

[4] The total drainage area of the main basins in the range, draining the main divide to the mountain front, at elevation $z_{th}$ used for the analysis. The calculation of $\theta$, $\Delta\mathcal{L}$ and $\Delta\chi$ is based on the drainage network defined by these basins, and the calculation of the aridity index statistics is based on a mask defined by the boundaries of these basins.

[5] Apparent steepness index based on the linear regression slope of the $\chi$-$z$ domain.

[6] The Aridity Index is calculated based on a mask defined by the main basins, based on Zomer et al. (2022).

[7] Number of paired flow pathways analyzed for $\Delta\mathcal{L}$ and $\Delta\chi$.

[8] Due to the extensive area of these ranges, the concavity index calculation and the extraction of the channel heads used for the analysis of $\Delta\mathcal{L}$ and $\Delta\chi$ were based on a trimmed drainage network after eliminating channels of Strahler order one.



**Cenral Mountain Range, Taiwan**

**Clan Alpine Mountains, Nevada**

**Daquing Shan, China**

**Finisterre Range, Papua New Guinea**

**Humboldt Range, Nevada**

**Inyo Mountains, California**

**Kammanassie Mountains, South Africa**

**Lüliang Mountains, China**

**Panamint Range, California**

**Sakhalin Mountains, Russia**





**Figure A1.** Relief and drainage networks of the 18 elongated mountain ranges used in the analysis of figures 2a and 6 in the main text.





## Appendix B: $\theta$ - $\Delta\mathcal{L}$ insensitivity to threshold drainage area in natural mountain ranges

Figure 1a shows the relation between $\theta$ and $\Delta\mathcal{L}$ for 18 elongated mountain ranges across the globe. The figure is based on drainage network extraction using a drainage area threshold of 145 pixels, corresponding to $\approx 1$ km$^2$. Figure B1 shows that when the drainage network is defined based on different thresholds, the distributions of $\theta$, $\Delta\mathcal{L}$, and $\Delta\chi$ varies, but the emerging
trends between these parameters are generally independent of the threshold drainage area.

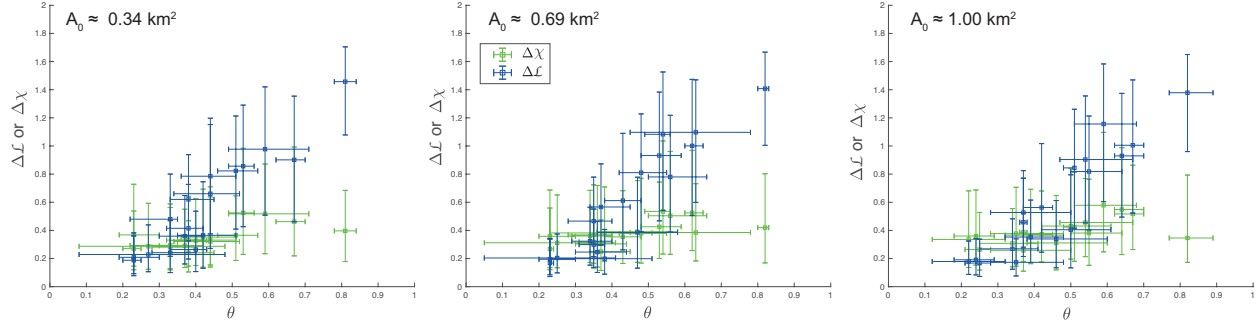

**Figure B1.** The effect of threshold drainage area, $A_0$ used for the extraction of the drainage network on the relations between $\Delta\mathcal{L}$ or $\Delta\chi$ and the concavity index, $\theta$. The choice of $A_0$ affects the calculated values, but the positive relation between $\Delta\mathcal{L}$ and $\theta$ and the less significant correlation between $\Delta\chi$ and $\theta$ are reproduced in all cases. The three panels show data for the same elongated ranges, except the Taiwanese central mountain range, which is omitted from the left panel because the concavity estimation with $A_0 \approx 0.34 km^2$ produced a bi-modal distribution of values.

## Appendix C: $\theta$ - $\Delta\mathcal{L}$ and the erodibility in the numerical DAC simulaitons

The erodibility coefficients used in the DAC simulations, as shown in Figure 2b, were varied along with the value of $\theta$ to maintain an approximately equal range relief (Willett, 2010). These values are listed in Table C1.

To ensure that the erodibility itself does not control $\Delta\mathcal{L}$, we repeat a subset of the simulations, where the concavity index
varies, and the erodibility is kept constant, resulting in numerical ranges of drastically different relief. Figure C1 shows the same trend as in figure 2b in the main text, indicating that this trend is independent of changing the erodibility index.

## Appendix D: The evolution of $\Delta\chi$ during OCN simulations

The main text shows the evolution of the normalized energy equivalent and $\Delta\mathcal{L}$ as functions of iteration number during the greedy OCN optimization application. Here, the behavior of $\Delta\chi$ is reviewed. Figure D1 shows that the energy minimization
process also decreases the median $\Delta\chi$. However, compared to the normalized energy equivalent and the $\Delta\mathcal{L}$ trends (Figures 4a and 4b in the main text), the $\Delta\chi$ decrease is more uniform across values of $\theta$, as also seen in figure 2c in the main text. Furthermore, the decrease in $\Delta\chi$ is non-monotonous.





**Table C1.** Concavity index and Erodibility in the DAC simulations [1]

| $\theta$ | $K[\text{L}^{(1-3m)}/\text{T}^{(1-m)}]$ |
|---|---|
| 0.1 | 9.2828d-4 |
| 0.2 | 1.5273d-4 |
| 0.3 | 2.5936d-5 |
| 0.4 | 4.562d-6 |
| 0.5 | 8.3312d-7 |
| 0.6 | 1.5811d-7 |
| 0.7 | 3.1152d-8 |
| 0.8 | 6.3577d-9 |
| 0.9 | 1.3392d-9 |

[1] Parameters used in the simulations depicted in figure 2b.

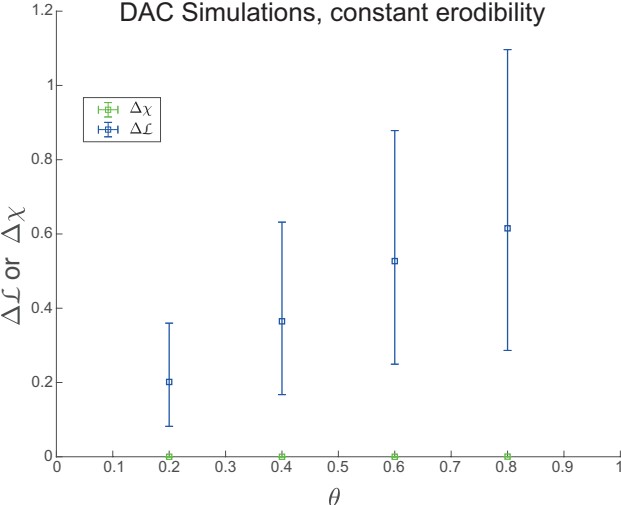

**Figure C1.** Relation between $\Delta\mathcal{L}$ (blue) and $\Delta\chi$ (green) and the concavity index, $\theta$ for a subject of $\theta$ values. Here, the concavity index, $K$ is maintained constant. Yet, the relations observed in figure 2b remain the same.





OCNs can be linked to drainage network evolution over topography (Banavar et al., 2001). Banavar et al. (2001) found that any edge flip resulting in a reduction of the network's total energy indicates that the flip occurred towards a steeper immediate
neighbor. While such a flip aligns with a decrease in $\Delta\chi$ between the pre- and post-flip channel heads, the non-monotonic trends of $\Delta\chi$ depicted in Figure D1 indicate that the median $\Delta\chi$ of the network, encompassing the entire model domain, does not necessarily decrease following an edge flip. This is likely because a single flip changes the drainage area distribution along the two affected basins, which can also change the $\Delta\chi$ across other channel head pairs.

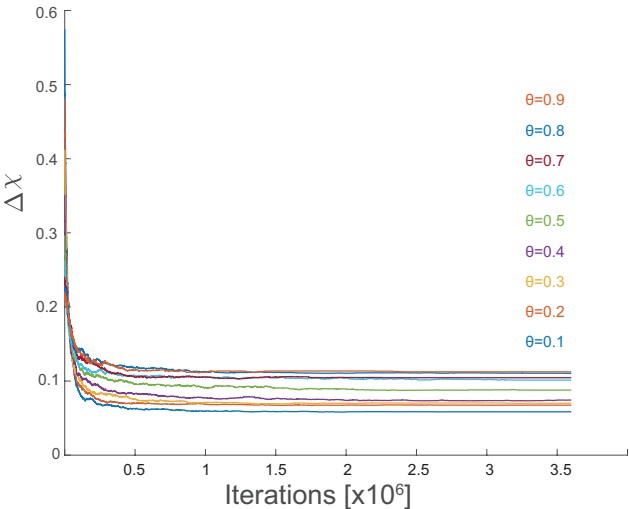

**Figure D1.** Evolution of median $\Delta\chi$ during the optimal channel network (OCN) simulations (see Method section in the main text) for different values of the concavity index $\theta$.

*Author contributions.* LG and ES conceptualized the project and designed the analysis. LG developed the code and executed the analysis.
ES contributed to the interpretation of the results and the definition of their implications. LG wrote the first draft and prepared the figures. LG and ES edited the text.

*Competing interests.* The authors declare that they have no conflict of interest.

*Acknowledgements.* This research was funded by the mutual program of the U.S. National Science Foundation and U.S.–Israel Binational Science Foundation (grant no. 1946253 (NSF) and 2019656 (BSF)) and by the Israel Science Foundation (ISF Grant 562/19). The authors
thank R. Hagbi for compiling the aridity index data used for producing figure 6.



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
