# Peer review of "Channel concavity controls plan-form complexity of branching drainage networks"

_EGUsphere, 2024_

## Referee Comment (RC2)

Paper Entitled:   Channel concavity controls plan-form complexity of branching drainage networks

Author:   L. Goren and E. Shelef

Manuscript no. : https://doi.org/10.5194/egusphere-2024-808

Recommendations:

This paper presents a framework to quantify channel networks' complexity based on the distribution of lengthwise asymmetry. Using both field observations and numerical simulations, the authors argue that the channel concavity index is the major control of planform complexity. The manuscript is interesting and written well. Given the data and the methodology employed, the article is relevant to the journal, however minor changes and clarifications ought to be made prior to its publication.

Below, I outline a few examples of specific points and clarifications:

- Page 15, Line 414: Do these computed or assumed $\theta$ values represent here single scaling regimes? For example, if computed from slope-area curve, $\theta$ value may be different for different range of scales for same topography and can vary with climate (see Hooshyar et al 2017). Also, smaller $\theta$ is usually associated with colluvial channels/sub-basins which tend to exhibit side-branching vs branching structures. Are these simple-looking configurations referred to such subbasins?
- How do topographies with multiple concavity index exhibit complexity? Are they more complex than topographies with single $\theta$? Are there certain range of scales that dominate network complexity?
- Line 215: Is there a dependence (linear or nonlinear) of $\theta$ on climate aridity? Looking at figure 6, it is not very apparent as the data show high variability. Zanardo et al 2013 showed that the c-value (expressing side-branching in topology, a measure of RN's topological complexity) increases with increasing precipitation, whereas Ranjbar et al 2020 showed that the network complexity, quantified via entropy, increases with increasing c-value. I wonder if the authors observe similar relation of $\theta$ with climate aridity- perhaps such a curve of $\theta$ vs AI may be useful.
- Line 246: Power-law (typo)
- Figure 4: Although not in terms of $\theta$, similar observations were made by AbedElmdoust et al 2016 (see their Figs 1 and 2).
- Line 80: How is the stationary defined? In X and Y only or in X, Y and Z as well?

Zanardo, S., I. Zaliapin, and E. Foufoula-Georgiou, Are American rivers Tokunaga self-similar? New results on fluvial network topology and its climatic dependence (2013), J. Geophys. Res. Earth Surf., 118, 166-183, doi:10.1029/2012JF002392.

Ranjbar S., A. Singh, and D. Wang (2020), Controls of the Topological Connectivity on the Structural and Functional Complexity of River Networks, Geophys. Res. Lett., https://doi.org/10.1029/2020GL087737.

Hooshyar, M., A. Singh, and D. Wang (2017), Hydrologic controls on junction angle of river networks, Water Resour. Res., 53, doi:10.1002/2016WR020267.

Abed-Elmdoust, A., M.-A. Miri, and A. Singh (2016), Reorganization of river networks under changing spatiotemporal precipitation patterns: An optimal channel network approach, Water Resour. Res., 52, doi:10.1002/2015WR018391.

---

## Author Comment (AC1)

**Dear Reviewers and editors,**

**We thank Dr. McNab and an anonymous reviewer for thoughtful and constructive comments.**

**Below, we list our replies to the points raised by the reviewer, emphasizing intended changes to the manuscript that specifically address the comments.**

**RC1 with replies**

This manuscript explores the origins of complexity in river networks. The authors make a persuasive case for the importance of river network complexity, which influences patterns surface runoff, erosion, sediment transport, relief generation, and the distribution of ecological niches. By analysis of natural landscapes, landscape evolution simulations, and the construction of energy-minimizing networks, the authors show convincingly that network complexity is closely associated with the concavity index of associated river long profiles. They interpret these patterns in terms of the global energy associated with a water flow through a given network planform. When concavity is low, slope changes slowly with increasing drainage area, so that parts of the network with high drainage areas (hence high discharges) are still relatively steep. This combination of high steepness and high discharge results in high energy expenditure, so that the network will evolve to minimize drainage areas, and hence towards a simple planform. When concavity is high, slope decreases more quickly with increasing drainage area, reducing the energy cost associated with higher drainage areas, and permitting more complex networks. With reference to further illustrative landscape evolution simulations, the authors discuss some interesting implications of this result, including the response of network planforms to changing climate and planform 'memory' of lithological structures that have long eroded away.

Overall I found the manuscript to be well argued, well written and well-illustrated. The results and discussion are convincing, novel, of interest to the general geomorphological community, and important for those specifically engaged in landscape evolution modelling. As such, I think the manuscript is certainly appropriate for publication in *ESurf* and could probably be accepted as it is. I do have some minor suggestions though which I would ask the authors to consider. These comments mostly concern: (a) descriptions of the methods used and (b) queries concerning some of the results and some suggestions for further discussion. I will give more details below, along with some technical corrections I found.

Lastly, I stress again that I consider these issues minor and that my comments are intended only to improve what I think is already a high-quality manuscript. I congratulate the authors on what I believe will make an excellent contribution to *ESurf*!

*Thank you for the supportive and thought-provoking comments and useful edit suggestions.*

I felt several of the method descriptions would benefit from an additional sentence or two giving the reader an idea of the concepts behind the chosen methods. In most cases the authors point to the original literature, but I think providing a small amount of additional explanation will help the reader follow what is being done and why, and also make clearer what the underlying assumptions are.

*We plan to submit a revised version with more detailed elaboration on the various methodological aspects of our study.*

For example, on lines 131-139, the authors describe how they measure the concavity index for the natural landscapes. They state that they use a 'disorder scheme', which 'asses the extent to which the elevation-based order of channel pixels aligns with their order by their χ value'. I think the basic idea here is that the points in the landscape with the same χ value should be at the same elevation – but I think it could be stated more clearly for the benefit of readers less familiar with such analysis. I think this approach also assumes uniform uplift across the domain, so that 'signals' are introduced to the long profiles only at base level – this assumption is maybe reasonable given the authors' choices of study areas, but could be stated explicitly (along with any other key assumptions behind the method).

*Conceptually, the disorder scheme explores the degree to which ordering pixels by elevation produces a similar order to ordering pixels by chi values. It allows for identifying a range of concavity indices that minimize the 'distance' between the elevation and chi ordering. The approach does not strictly assume a spatially uniform uplift rate, but it assumes sufficiently small spatial variations such that the pixel ordering by chi and elevation correspond to one another. These ideas will be expanded upon in the revised manuscript.*

Similarly, on lines 193-207, the authors describe the method for computing optimal channel networks. They introduce their 'greedy' algorithm, and contrast it to an 'annealing' algorithm that involves a 'temperature dependent probability'. These terms do not mean much to me, and suspect not to the general reader either. So, if the distinction is important, I think it would be useful to explain the it in simpler terms here, if possible. I think that would be particularly important if the greedy algorithm the authors employ is new, which is ambiguous in the current description.

*'Simulated annealing' algorithms enables the system to exit local minima (in energy) by prescribing a time-dependent (in practice, a temperature dependent) probabilistic*

*threshold for accepting energetically unfavorable edge flips. This approach allows the system to exit local minima and explore the space more thoroughly, yet the probabilistic component makes direct interpretation more difficult.*

*The 'greedy' approach takes a simpler path by allowing only edge flips that decrease the total energy, leading to faster convergence to a local minimum. This method follows the work of Rodríguez-Iturbe et al. 1992 (a reference that will be added to the revised version of the manuscript) and it does not involve any computational complexities. It is similar to the 'annealing' approach when the probability for accepting energetically unfavorable edge flips is set to zero. As all simulations start with the exact same network configuration, the greedy approach enables direct comparison between resulting networks of different concavity, whereas the probabilistic component of the annealing approach introduces significant stochasticity to the topology of the initial configuration that makes it difficult to compare given the context of this study. This concept is explained in lines 198-199 in the original submission. Since networks are not guaranteed to converge to a global optimum even under the 'annealing' approach, the networks produced by the 'greedy' approach are not expected to be statistically different from those produced with the 'annealing' approach.*

*Our current thinking is not to expand on these issues within the manuscript because we consider them methodological details. Interested readers are referred to previous literature. They can also read this discussion and download and use the accompanying code, which is available on GitHub.*

Finally, a more minor example, on lines 147-150, the authors describe an elevation correction, but it is not explained why this correction is needed. Indeed, maybe I missed something, but I did not notice anywhere in the later analysis that elevation is used, so it is still unclear to me why this correction is applied.

*We note that the correction was introduced to the chi and L values, not to the elevation per se. The idea is that chi gradients and L gradients should be compared between channel heads of equal elevation across the divide. In cases where the extracted natural and numerical channel heads are not of equal elevation, the correction calculates chi and L values for a fictive channel head, corresponding to the elevation of the higher channel head of the pair, while assuming uniform steepness ($K_s$) along the flow path. This motivation will be better explained in the revised manuscript.*

*Queries / discussion points*

Below I list some queries that came up for me while reading the manuscript. I do not think any of these points represent significant issues that necessarily need to be

addressed in a revised manuscript. But they could warrant comment or further discussion, if of interest to the authors.

Throughout the manuscript I wondered about the potential importance of drainage density/the relative importance of hillslope vs. fluvial processes for the network complexity as defined here ($\Delta L$). Intuitively I would have thought that a landscape with fewer channels would be more likely have a simple network structure than one with a dense channel network. In landscape evolution models, the density depends on model parameters like erodibility, concavity (i.e. $m$ & $n$), and hillslope diffusivity (see e.g. Perron et al., 2008, *JGR*). (A related question: do the DAC models shown here have a hillslope component? If so, it is not mentioned in the method description.) Theodoratos et al. (2018, *ESurf*) define a lengthscale, $l_c$ (their equation A8), which depends on $m$ and describes the relative importance of hillslope vs. fluvial processes: if $l_c$ is small, fluvial processes dominate and a high channel density will develop; if $l_c$ is large, hillslope process dominate and a more diffuse landscape develops. So, I wonder if an alernative explanation for the dependence of complexity on $\theta$ would be that we are looking at landscapes with different relative contributions from hillslope and fluvial processes? Since there are no hillslopes in the energy-minimising network simulations, I think the authors' interpretation of the results in terms of energy minimisation are probably sound. But perhaps this would be a useful/interesting point to test/discuss. For example, an additional set of experiments could vary $m$ but also an additional parameter (erodibility or hillslope diffusivity) to keep $l_c$ constant. Alternatively, hillslope diffusivity could be varied while $m$ is fixed to test if that has any influence on $\Delta L$.

***In the DAC LEM, all grid nodes are fluvial, and the hillslope component is generally solved analytically at the sub-grid resolution. In the version we use here, the hillslope component is suppressed. Regardless of how hillslopes are treated in DAC, the drainage density is controlled based on the typical grid spacing, so there were no differences in drainage density between DAC simulations of different concavity. Consequently, the association between concavity and complexity in both the DAC simulations and the OCN simulations is independent of the drainage density, which was maintained constant. We will emphasize these crucial issues in the revised version.***

***When analyzing the natural elongated mountain ranges, we extracted the drainage network based on a threshold drainage area, Ac, (which depends on lc ), consequently, the extracted drainage networks are expected to have an approximately similar density. We tested the robustness of the results against different assumptions for channelization threshold drainage area, Ac (which depends on lc in Theodoratos et al. 2018). We found that the association between***

*concavity and complexity is independent of Ac (Appendix B). This again shows that the correlations we identified are likely independent of the drainage density. We agree that further tests should be done to quantify the contribution of drainage density to complexity but prefer to reserve this exploration for future studies.*

During the discussion of Hack's law (Section 5.1), the authors show that the observed relationship between θ and Δ*L* can be explained by some variation in the Hack's law parameters *k* and *h*. It is interesting that the data seem to fall roughly into a specific variation band (maybe $k_i - k_j$ = c. 0.2, $h_i - h_j$ = c. 0.1, the specific band varying a bit with theta). Does this result point to some characteristic local variability in Hack's law parameters in natural networks, which could potentially be measured? And could it then form a future test of the authors' interpretations? If so, it might be worth commenting on. The authors quote some references on variability in Hack's law parameters but it is not clear whether these numbers apply on a global or local scale (the latter being the relevant one here).

*The range of Hack's coefficients and exponents we cite is based on global and regional studies. We are not aware of any studies that have examined variations of Hack's parameters locally, or more specifically, across basins sharing a divide. We agree this is a valuable area for exploration, it is likely beyond the scope of the current manuscript. Including it might shift the focus and dilute the main message of our study.*

Lastly, there are two definitions of a 'stable' network used in the manuscript – one says that χ should be equal across drainage divides, the other that the network is 'optimal' from an energy point of view. These two definitions seem to point in the same direction in terms of the relationship between concavity and complexity. But are they generally consistent with one another? i.e. do networks that satisfies one condition always satisfy the other? It may be worth noting and comparing to two definitions explicitly somewhere.

*This point is explored in Appendix D and Figure D1. Our analysis shows that energy minimization also decreases the median Delta chi values across the divide. However, unlike the monotonic energy reduction forced by the greedy OCN algorithm, the Delta chi reduction is not monotonic. We note that Banavar et al. (2001), cited in Appendix D, theoretically explored the relationship between these two definitions of stability (they did this before chi was formally defined and used elevation along the network instead).*

**Technical corrections**

Throughout the manuscript the authors hyphenate "plan-form", which I have not seen before. I know "planform" as a word in itself, which I think could be safely used here. But perhaps the authors have their reasons for this choice.

*We will switch to 'planform' in the revised version.*

L49. By "inclination" do you just mean "slope"?

*Yes. We will change to slope in the revised version.*

L67-68. "An alternative effect relates the concavity index to the lengthwise asymmetry between paired flow pathways that diverge from a single divide and rejoin at a junction or a common base level." This phrase comes out of the blue a bit and on first reading I did not understand it. I see now how it relates to the definition of complexity used later ($\Delta L$). But perhaps it is not necessary not bring that up at this stage in the introduction.

*In the revised version, we plan to rephrase and expand on this idea to introduce it more gradually and make it feel less abrupt.*

L140. "… with distance from divide endpoints exceeding 1000." What are the units here? Pixels? Metres?

*The distance is measured with the DEM spatial dimensions (Scherler, D. and Schwanghart, W., 2020. Drainage divide networks–Part 1: Identification and ordering in digital elevation models. Earth Surface Dynamics, 8(2), pp.245-259.) This will be clarified in the revised version.*

L152. In what sense is the DAC model "process based"? It solves the stream power model which I would consider empirical/phenomenological.

*This could be a matter of perspective. In our view, the stream power model is not purely empirical or phenomenological. While the exponents and coefficients are indeed empirical/phenomenological, its form is rooted in the physical intuition that the erosion rate scales with the shear stress (or (unit) power) of the flowing water (which could be seen as a proxy to detachment capacity and availability of erosion "tools"). Therefore, an assumed process (shear stress induces erosion) is involved, though the details of the process are not specified. In the current study, it is contrasted with the OCN, which is an energy optimization model.*

L156. The equivalence of the stream power model and Flint's law is only true at steady state; the statement here should be qualified.

*They are identical if Flint's law is defined at a specific point, meaning $K_s$ is not necessarily spatially invariant. Assuming E and Kare fully known, the stream power model predicts $S(x,t) = (E(x,t)/K(x,t))^{1/n} A(x,t)^{theta}$, which matches the form of Flint's law when the coefficient $K_s$ is not assumed constant. Here, E is not necessarily equal to U and is expected to vary along the channel based on the spatio-temporal history of U. Despite this relation, we will likely omit this comparison in the revised version to avoid confusion.*

L205-207. "… $\Delta L$ and $\Delta \chi$ are calculated only for neighbouring channel head pairs that drain to different outlets (boundary nodes)." I am a bit confused by this part. Outlets/boundary nodes sounds to me like the edge of the model domain at base level. But then if the channels drain to different outlets, ΔL and Δχ can't be calculated. So I assume you are talking about the immediate downstream nodes? Perhaps this sentence could be rephrased to clarify.

*Yes, outlets or boundary nodes refer to the edges of the domain, which hasa common, same-elevation base level. Throughout the analysis, Delta L and Delta chi are calculated based on distances to the common junction or common base level—the domain edge—allowing us to also calculate Delta L and Delta chi for divides that separate different basins.*

L278. "Exploiting" is quite strong for me and implies agency on the part of the network which I don't think is quite right – maybe "facilitated by small variations in Hack's exponent and coefficient"?

*This rephrase suggestion will be adopted in the revised version.*

L308. "… an exceptionally simple geometry that differs significantly from any random network." I think this statement is too strong: simple geometries can arise from random processes, though they may be unlikely. Perhaps something like: "… an exceptionally simple geometry that is extremely unlikely to arise from any random process …".

*We weren't sure about this issue and used ChatGPT to get a sense of the relevant probabilities. The number of random networks scales as (number of nodes)! * (number of neighbors)^(number of nodes), which for the current application is 12000! * 7^12000. The general form of the formula is our intuition, not ChatGPT's. The number 7 represents the number of neighbors minus the donor, assuming only a single donor. While there are cases with multiple donors, a node could also be the outlet. This can be expressed with 53,895 digits (ChatGPT4o). If we very conservatively assume that low complexity networks are generated by directing the flow to one of three neighbors that are closer to the base level, then the number of such networks is 12000! * 3^12000 (our intuition), which needs 49,478 digits to be expressed. The probability of randomly*

*selecting a low complexity network is the ratio of the two, which is 10^-4416 (ChatGPT4o). When I asked ChatGPT4o to supply a real-world analogy for how small this value is, one of her replies was: The probability of randomly selecting a specific atom out of all the atoms in the observable universe is around 10^−80. This number is incomparably larger than 10^−4416. In light of this calculation, 'differs significantly' is probably not too strong. Note that our estimation of the number of general and low-complexity networks represents an extreme overestimation because the current count allows loops and does not ensure that all trees (basins) in the forest (network) are rooted in a boundary/base-level node.*

L316. I think an "a" is missing between "reveals" and "monotonous".

*Will be corrected in the revised version*

Figure C1: I think the "concavity index, $K$" should be the "erodibility, $K$", or similar.

*Will be corrected in the revised version*

**RC2 with replies**

This paper presents a framework to quantify channel networks' complexity based on the distribution of lengthwise asymmetry. Using both field observations and numerical simulations, the authors argue that the channel concavity index is the major control of planform complexity. The manuscript is interesting and written well. Given the data and the methodology employed, the article is relevant to the journal, however minor changes and clarifications ought to be made prior to its publication.

*Thank you for the constructive and helpful comments, and for the highly relevant references.*

Below, I outline a few examples of specific points and clarifications:

- Page 15, Line 414: Do these computed or assumed theta values represent here single scaling regimes? For example, if computed from slope-area curve, theta value may be different for different range of scales for same topography and can vary with climate (see Hooshyar et al 2017). Also, smaller theta is usually associated with colluvial channels/sub-basins which tend to exhibit side- branching vs branching structures. Are these simple-looking configurations referred to such subbasins?

*The inferred theta values, which were extracted using the disorder method (not slope-area), are assumed to represent basins larger than 10^5 m^2. Similarly, in the simulations, we applied a single theta value for the entire landscape, as all DAC nodes represent fluvial channels. We agree with the reviewer that this is a simplifying view because the concavity index could change across the landscape, in between, and within basins. In the revised version, we will make sure to make this assumption explicit.*

*Thank you for pointing out the missing relevant reference to Hooshyar et al. (2017). Regarding this reference, given that the threshold drainage area we used for extracting the drainage network from natural landscapes is > $10^5$ m², it is larger than the threshold found for the transition in concavity in Hooshyar et al. (2017). Additionally, the insensitivity of our results to the threshold drainage area, as depicted in Figure B1, likely means that the inferred concavity index is not significantly affected by colluvial/debris flow basins.*

*Several additional studies have shown that theta could be climate- and process-dependent (with a strong drainage area control). The implications of this are explored in Sections 5.3 and 5.4.1.*

How do topographies with multiple concavity index exhibit complexity? Are they more complex than topographies with single theta? Are there certain range of scales that dominate network complexity?

*Addressing this point requires a smaller-scale analysis of natural landscapes (e.g., exploring situations where space-dependent theta can better fit the drainage network), together with more complex landscape evolution simulations, where theta is set to vary in space. These are interesting and relevant tests. However, they are beyond the scope of our current analysis. We hope to explore them in future studies.*

*If we have to guess, the complexity will likely depend on the spatial variability in theta. For example, if theta changes with drainage area—transitioning from fluvial-dominated high theta in large drainage areas to debris flow-dominated low theta in small areas—then the median landscape-scale complexity would likely represent an intermediate theta value. In this case, when a pair of channel heads and their junction lies in the small drainage area domain, they do not 'feel' the high concavity and will show high lengthwise symmetry. When the junction is closer to the base level, with high drainage area, their path would 'feel' the high concavity and their symmetry could be lower. Overall, the median would be somewhere in between. If, however, theta varies between neighboring basins or sub-basins, then the diverging paths from any divide would be characterized by different theta values, one of which is potentially high and potentially with a tortuous path. This is similar to the high theta configuration. In this*

*case, the landscape-scale complexity may exhibit large values, effectively corresponding to the greater theta.*

*This discussion further points to the idea that complexity could be scale-dependent, which is another topic for future exploration.*

Line 215: Is there a dependence (linear or nonlinear) of theta on climate aridity? Looking at figure 6, it is not very apparent as the data show high variability. Zanardo et al 2013 showed that the c- value (expressing side-branching in topology, a measure of RN's topological complexity) increases with increasing precipitation, whereas Ranjbar et al 2020 showed that the network complexity, quantified via entropy, increases with increasing c-value. I wonder if the authors observe similar relation of theta with climate aridity- perhaps such a curve of theta vs AI may be useful.

*This is indeed an intriguing question. The figure below shows the relationship between theta and AI for the 18 elongated mountain ranges we studied.*

[Figure]

*Although some trend is apparent, the Spearman's rank correlation coefficient for this correlation is 0.37, and the correlation is not statistically significant (alpha=0.05), with a P-value of 0.13. The data shown in Figure 6 of the relationship between complexity and the aridity index is statistically significant.*

*Previous studies, which used much larger datasets, were able to identify a correlation between theta and climate (e.g., Zaprowski et al., 2005; Chen et al., 2019; Getraer and Maloof, 2021). A possible interpretation for the statistical insignificance of the relationship between theta and AI in our data (above figure) versus the statistical significance of the relationship between AI and complexity (Figure 6) is that the measurement of channel concavity in natural settings might encompass several environmental factors, including variations in tectonics (i.e., Seybold et al., 2021) and rock types, introducing noise into the relationship between concavity and aridity. We will likely add the figure above to the manuscript appendix, while addressing in the main text the relationships between theta and AI versus complexity and AI.*

*In contrast, our analyses, based on Hack's law and OCN theory, demonstrate that network complexity is contingent on the channel formative concavity index, which is expected to depend on hydrologic conditions (i.e., Whipple and Tucker, 1999) and, therefore, is expected to exhibit a more robust correlation with climate.*

*Thank you for pointing out the two relevant references, Zanardo et al. (2013) and Ranjbar et al. (2020). We will make sure to include them in a short introductory summary about approaches for quantifying network complexity.*

Line 246: Power-law (typo)

*Will be corrected in the revised version.*

Figure 4: Although not in terms of theta, similar observations were made by Abed Elmdoust et al 2016 (see their Figs 1 and 2).

*Thanks for pointing out the relevant (and nice!, we enjoy OCN very much) manuscript of Abed Elmdoust et al. 2016. We will make sure to include it in the revised version.*

Line 80: How is the stationary defined? In X and Y only or in X, Y and Z as well?

*X and Y only. Changes in divide height could stem from symmetric (across divide) hillslope processes, which do not affect the lengthwise asymmetry.*